# Persistent cell migration emerges from a coupling between protrusion dynamics and polarized trafficking

**Kotryna Vaidžiulytė[1,2,3], Anne-Sophie Macé[4], Aude Battistella[1], William Beng[1], Kristine Schauer[2,5]*, Mathieu Coppey[1]***

[1]Laboratoire Physico Chimie Curie, Institut Curie, PSL Research University, Sorbonne Université, Paris, France; [2]Cell Biology and Cancer Unit, Institut Curie, PSL Research University, Sorbonne University, Paris, France; [3]Faculty of Science and Engineering, Sorbonne Université, Paris, France; [4]Institut Curie, PSL Research University, Paris, France; [5]Tumor Cell Dynamics Unit, Gustave Roussy Institute, Université Paris-Saclay, Villejuif, France

**Abstract** Migrating cells present a variety of paths, from random to highly directional ones. While random movement can be explained by basal intrinsic activity, persistent movement requires stable polarization. Here, we quantitatively address emergence of persistent migration in (hTERT)–immortalizedRPE1 (retinal pigment epithelial) cells over long timescales. By live cell imaging and dynamic micropatterning, we demonstrate that the Nucleus-Golgi axis aligns with direction of migration leading to efficient cell movement. We show that polarized trafficking is directed toward protrusions with a 20-min delay, and that migration becomes random after disrupting internal cell organization. Eventually, we prove that localized optogenetic Cdc42 activation orients the Nucleus-Golgi axis. Our work suggests that polarized trafficking stabilizes the protrusive activity of the cell, while protrusive activity orients this polarity axis, leading to persistent cell migration. Using a minimal physical model, we show that this feedback is sufficient to recapitulate the quantitative properties of cell migration in the timescale of hours.

*For correspondence:
kristine.schauer@gustaveroussy.fr (KS);
mathieu.coppey@curie.fr (MC)

**Competing interest:** The authors declare that no competing interests exist.

## Editor's evaluation

It has previously been suspected that secretion supports cell migration in human cells. This study proposes a physical model and offers various results that elegantly link the activation of a small GTPAse at the leading edge with the re-organisation of the secretory pathway, creating a feedback loop that allows persistence of direction. Hopefully, the simple physical model will serve as a foundation to include more regulatory loops in the conceptualisation of cell migration.

## Introduction

Cell migration is involved in many processes such as development, invasion, wound healing, or immune response (*Vicente-Manzanares and Horwitz, 2011*). There is an impressive variety of modalities by which cells migrate, including mesenchymal or amoeboid type of movement for which single cells or a group of cells (*Shellard and Mayor, 2019*) use different propulsive forces for displacement (*Othmer, 2018*). Regardless of the propulsive force or single/collective mode of migration, cells polarize to move (*Rappel and Edelstein-Keshet, 2017*). This is characterized by an asymmetric shape and distribution of proteins, organelles, and lipids, as well as differential activities at the two extreme sides of the cell (*Vaidžiulytė et al., 2019*). This polarity allows cells to spatially segregate propulsive and

contractile forces in order to move their body forward. In the context of mesenchymal cell migration, the polarity axis of cells is specified by a protruding front and a retracting back (*Ebnet, 2015*; *Ridley et al., 2003*). On the contrary, when cells are not polarized, they present several protruding regions along their contour and barely move (*Petrie et al., 2009*). Several mechanisms have been proposed to explain the long range coordination of front and back activities, from reaction-diffusion of signaling molecules (*Jilkine et al., 2007*), cytoskeleton template dynamics (*Gan et al., 2016*; *Maiuri et al., 2015*; *Prentice-Mott et al., 2016*; *Wang et al., 2013*), mechanical signals such as membrane tension (*Houk et al., 2012*), to contractility (*Cramer, 2013*; *Schuster et al., 2016*; *Vicente-Manzanares et al., 2011*; *Yam et al., 2007*). Eventually, numerous studies have highlighted the role of retrograde trafficking (*Shafaq-Zadah et al., 2016*) and directed secretion from the Golgi complex in sustaining persistent migration (*Hao et al., 2020*; *Yadav and Linstedt, 2011*; *Yadav et al., 2009*). However, it is not completely understood how these different mechanisms can be combined and what are their respective roles in allowing cells to maintain a stable polarity while migrating.

In the case of mesenchymal migration, cells move thanks to the sum of local protrusive activity (*Yamao et al., 2015*), and persistent migration relies on lamellipodial persistence (*Krause and Gautreau, 2014*). Protrusions are initiated and controlled by the small RhoGTPases (*Jaffe and Hall, 2005*; *Lawson and Ridley, 2018*). These signaling proteins are engaged in spatiotemporal patterns of activity (*Machacek et al., 2009*; *Fritz and Pertz, 2016*; *Pertz, 2010*), thanks to a large set of activators and deactivators, GEFs (Guanine nucleotide exchange factors) and GAPs (GTPase-activating proteins) (*Bos et al., 2007*; *Müller et al., 2020*). Among the RhoGTPases, Cdc42 has been recognized to be integrated into an excitable signaling network that can spontaneously polarize (*Yang et al., 2016*). Cdc42 crosstalks with polarity proteins (*Etienne-Manneville, 2008*; *Iden and Collard, 2008*) and with the cytoskeleton (*Bear and Haugh, 2014*). Notably, persistently migrating mesenchymal cells present a sustained and polarized internal organization, which can be viewed as an 'internal compass'. This compass corresponds to the polarity axis that can be represented by the axis from the nucleus to the centrosome or the associated Golgi complex (*Elric and Etienne-Manneville, 2014*; *Luxton and Gundersen, 2011*). In wound scratch assay, the Golgi complex reorients in front of the nucleus (*Etienne-Manneville, 2006*). Similarly, the centrosome reorients toward the leading edge during EMT (Epithelial-Mesenchymal Transition) (*Burute et al., 2017*). In other studies, the investigators have reported that the Golgi does not align with direction of migration at all (*Uetrecht and Bear, 2009*) or tends to be behind the nucleus when cells are studied on adhesive 1D lines (*Pouthas et al., 2008*). Thus, the role of the Golgi positioning and the internal compass in persistent migration remains to be clarified.

Based on pioneering work in yeast (reviewed in *Chiou et al., 2017*), cell polarity could be considered as an emergent property based on the coupling of high-level cellular functions, rather than being attributed to one specific pathway or to one single 'culprit' protein (*Vaidžiulytė et al., 2019*). Similarly, the emergence of persistency in cell migration could also rely on the coupling of high-level cellular functions. In the present work, we tested if the coupling between protrusion dynamics and internal cell polarity is present in mesenchymal cells and if this coupling could be sufficient to maintain persistent cell migration. For this, we quantified and manipulated the two subcellular functions described above at short and long timescales. Our experimental results were integrated into a minimal physical model that recapitulates the emergence of persistency from this coupling.

## Results

### Freely migrating RPE1 cells persistently protrude in front of the Golgi

First, we assessed the coupling between the internal polarity axis and cell protruding activity during persistent cell migration. We chose RPE1 cells which are known to have a reproducible internal organization (*Schauer et al., 2010*) and move persistently (*Maiuri et al., 2012*). To quantify the orientation of the internal polarity axis of the cell while migrating, we generated stable cell lines, with fluorescently labeled Golgi complex and nucleus. Rab6A fused to a GFP tag was overexpressed to follow the Golgi, and the nucleus was stained with Hoechst 33,342 (see Materials and methods). Cell contours were segmented in live by expressing an iRFP-fluorescent reporter anchored to the plasma membrane by a myristoylation motif. The live segmentation was employed to move the stage accordingly to the cell movement in order to keep the cell in the field of view (sup *Figure 1A* and

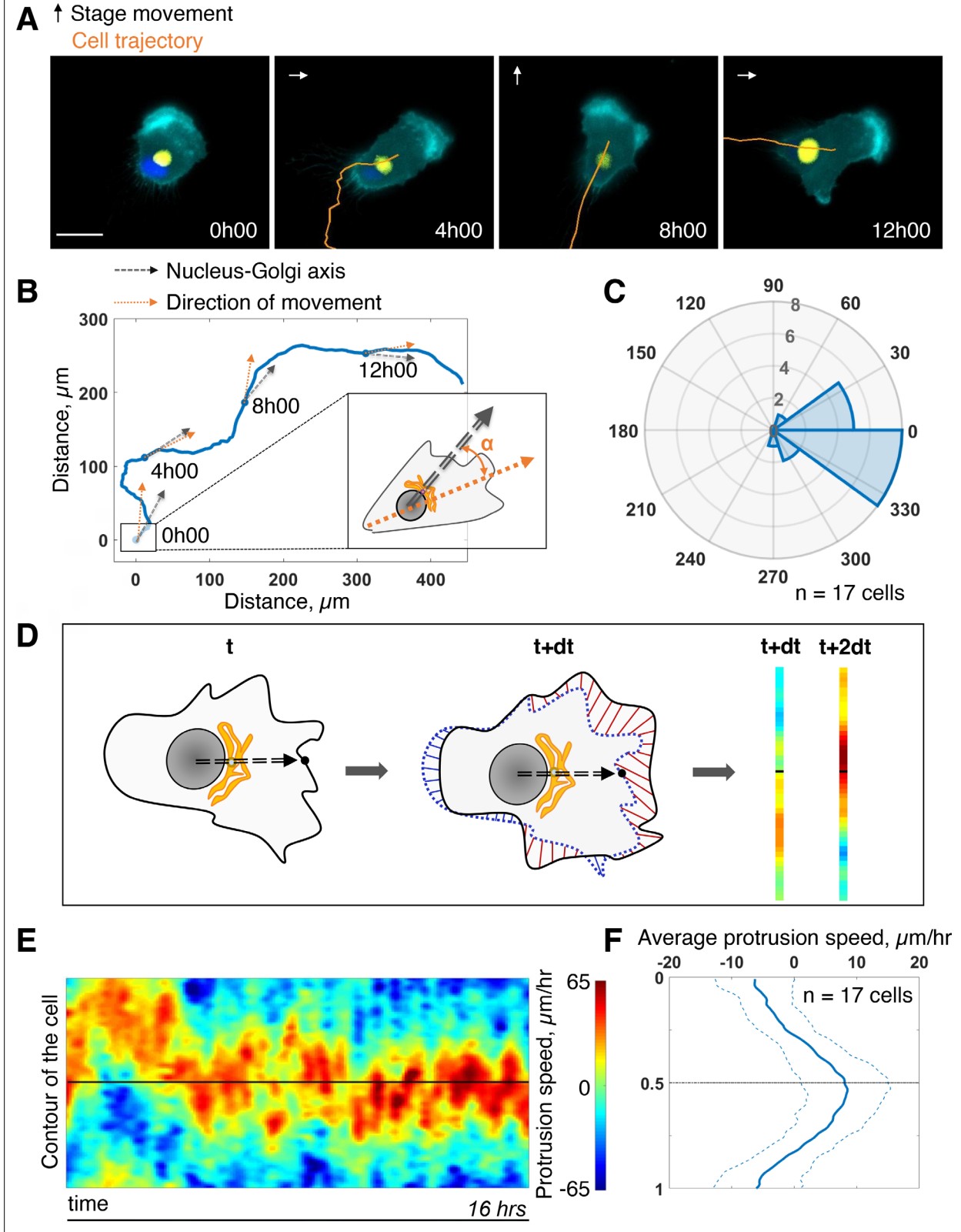

**Figure 1.** Persistent protrusions form in front of the Golgi complex. (**A**) Snapshots of a representative migrating RPE1 cell at different timepoints tracked with a feedback routine in which the microscope stage follows a migrating cell (*Figure 1—figure supplement 1*) for 16 hr (cyan: myr-iRFP, yellow: GFP-Rab6A, blue: Hoechst 33342, trajectory overlaid in orange, microscope stage movement represented by an arrow, scale bar – 20 μm). (**B**) Full trajectory of a representative cell shown in (**A**) with Nucleus-Golgi (black dashed arrow) and direction of movement (orange dashed arrow) axes overlaid. (**C**) Polar

*Figure 1 continued on next page*

*Figure 1 continued*

histogram representing the averaged angle between Nucleus-Golgi axis and direction of movement (n = 17 cells). (**D**) Explanatory sketch of how a morphodynamic map of cell shape changes is computed. The contour of the cell is extracted and compared between frames and stretched out to a line representation, where the distance traveled by a point in the contour is represented (red color meaning protrusion, blue – retraction, black dashed arrow – Nucleus-Golgi axis). (**E**) Morphodynamic map of a representative cell (all maps in *Figure 3—figure supplement 2*) recentered to Nucleus-Golgi axis (black). X-axis represents time and Y-axis represents cell contour. (**F**) Average protrusion speed over time (n = 17 cells, dashed blue line - SD). X-axis represents average protrusion speed, and Y- axis represents cell contour with the midline corresponding to (**E**). Data used for **C** and **F** and related scripts can be found in *Figure 1—source data 1*.

The online version of this article includes the following video, source data, and figure supplement(s) for figure 1:

**Source data 1.** Data and analysis scripts with explanations for *Figure 1* and its supplements.

**Figure supplement 1.** Feedback routine for moving the microscope stage (field of view) to follow a migrating cell.

**Figure 1—video 1.** Human telomerase reverse transcriptase (hTERT)–immortalized RPE1 cell freely moving on a fibronectin covered coverslip followed by a moving microscope stage.

https://elifesciences.org/articles/69229/figures#fig1video1

Materials and methods). This experimental strategy let us image cells with high spatial resolution for up to 16 hr with a 5-min temporal resolution (*Figure 1A* and *Figure 1—video 1*). For each timepoint, we quantified the direction of movement by measuring the displacement of the center of mass of the cell from the segmented images (orange arrow *Figure 1B*). We quantified the direction of the internal polarity axis by taking the vector joining the centers of mass of the nucleus to the Golgi (black arrow *Figure 1B*). We then computed the angular difference between the two vectors and averaged it over all timepoints and over 17 cells. The distribution of the angular difference is sharply pointing toward zero (–10° ±33°, *Figure 1C*), showing that there is a clear alignment of the Nucleus-Golgi axis with the direction of migration in RPE1 cells.

Next, we computed morphodynamic maps of the cell contour, which allows the visualization of protruding activity over time (*Machacek et al., 2009*). We computed these maps by measuring the displacement of the cell contour between two consecutive timepoints and by color coding the displacement, from blue (retraction) to red (protrusion) (*Figure 1D*, and Materials and methods). All displacements along the cell contour (y-axis) were plotted as a function of time. Using the direction of movement as reference (midline on the y-axis) throughout the movement, we found that the protrusive activity is perfectly aligned with the direction of movement (sup *Figure 1C and D*). This showed that cell migration is indeed driven by protrusions. When the Nucleus-Golgi axis is used as reference (midline on the y-axis), the protrusive activity appeared to align well with this axis (*Figure 1E*). Averaging over time and over cells, there is indeed a sustained protrusive activity in front of the Golgi, whose speed is significantly higher than throughout the cell (*Figure 1F*). These results demonstrate that there is a strong correlation between the direction of protruding activity driving cell movement and the orientation of the polarity axis in RPE1 cells indicating a strong coupling of these activities in freely migrating RPE1 cells.

## The Nucleus-Golgi axis does not predict the direction of migration, but aligns when cells move effectively

The correlation we observed does not imply a causal role of the internal polarity axis in driving the persistence of cell migration, because the Nucleus-Golgi axis may follow the direction of migration in a passive manner as a byproduct of cell morphological changes. Thus, we assessed its role by testing if cells start to move in a preferential direction along the given internal polarity axis (*Figure 2A*). For this, we employed the dynamic micropattern technique (*van Dongen et al., 2013*) that allows to release cells from a pattern. Cells are initially plated on round adhesive micropatterns coated with fibronectin surrounded by a repulsive PLL-PEG (poly(L-lysine)-poly(ethylene glycol)) coating. And 5 hr after plating, migration is initiated by adding BCN-RGD (bicyclo[6.1.0]nonyne-Arginine-Glycine-Aspartate) that renders the whole surface adhesive. Since the pattern is isotropic, there are no external cues to orient cell escape. We monitored cell movement by tracking the nucleus center of mass, and Nucleus-Golgi axis for 36 cells (*Figure 2B*, *Figure 2—figure supplement 1*, and *Figure 2—video 1*). During a first phase of ~5 hr cells remain on the pattern, and during a second phase they start to move out of it (*Figure 2—figure supplement 2A*). When we compared the orientation of the Nucleus-Golgi axis

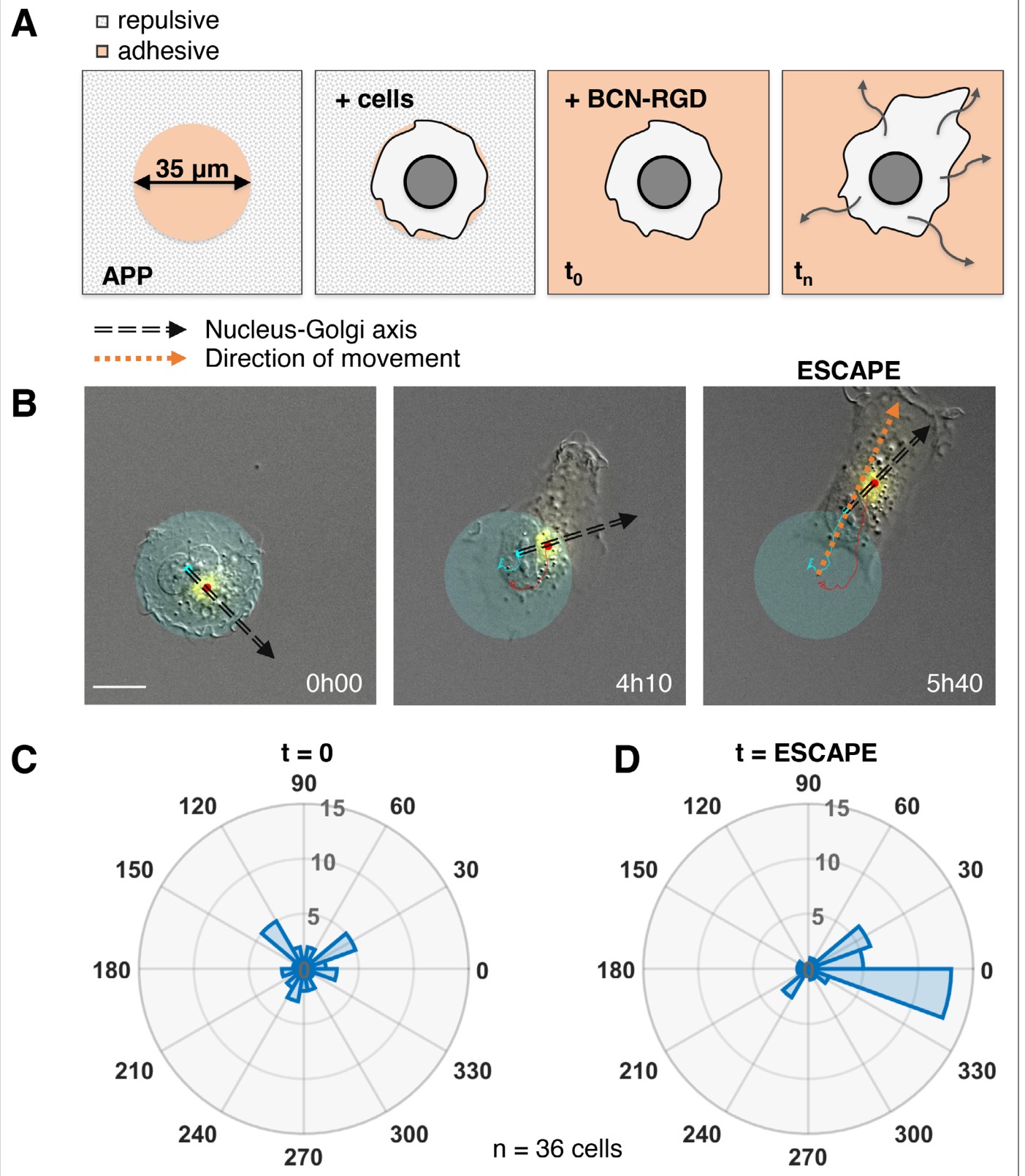

**Figure 2.** Nucleus-Golgi axis and direction of movement align when a cell starts moving. (**A**) Scheme of the dynamic micropatterns experimental design that is used to study the initiation of cell movement. A cell is confined on a round fibronectin pattern and after the addition of BCN-RGD is enabled to move outside and 'escape' the pattern ('escape' is defined to be the moment when the center of the cell nucleus is leaving the area of the pattern). (**B**) Representative RPE1 cell 'escaping' the pattern (transparent cyan: pattern, cyan dot: nucleus centroid, yellow: GFP-Rab6A, red dot: Golgi centroid, black

*Figure 2 continued on next page*

*Figure 2 continued*

dashed line: Nucleus-Golgi axis, orange dashed line: direction of movement, scale bar – 20 µm). (**C–D**) Polar histograms representing the angle between Nucleus-Golgi axis at the beginning of experiment (t = 0) (**C**) or at the time of 'escape' (t = ESCAPE) (**D**) and direction of movement when the cell moves out of the pattern (n = 36 cells). Data used for **C** and **D** and related scripts can be found in *Figure 2—source data 1*.

The online version of this article includes the following video, source data, and figure supplement(s) for figure 2:

**Source data 1.** Data and analysis scripts with explanations for *Figure 2* and its supplements.

**Figure supplement 1.** Detailed analysis of Nucleus-Golgi axis movement when a cell is escaping the pattern.

**Figure supplement 2.** Detailed analysis of Nucleus-Golgi axis movement when a cell is escaping the pattern.

**Figure 2—video 1.** RPE1 cell 'escaping' the pattern.

https://elifesciences.org/articles/69229/figures#fig2video1

at t = 0 (addition of BCN-RGD) with the direction of escape, we found no correlation (*Figure 2C*). This result shows that the direction of escape is independent from the initial positioning of the Golgi, as previously suggested in the literature (*Chen et al., 2013*). However, we found a clear correlation between the direction of escape and orientation of the Nucleus-Golgi axis at the time of escape (*Figure 2D*). A detailed temporal analysis (*Figure 2—figure supplement 2B*) showed that both the Nucleus-Golgi axis and cell direction of motion start to align about 2 hr before the escape. Our analysis indicates that they are concomitantly required to initiate effective migration.

## Disruption of microtubule dynamics abolishes persistence of migration on long timescales

Next, we addressed the role of the internal polarity axis on sustaining protrusion dynamics. Since microtubules (MTs) are known to play a major role in cell internal organization, we perturbed MT dynamics using low doses of Nocodazole (NZ), namely, 0.1 µM. This dose was sufficient to perturb MT dynamics without full disruption of MT network and did not impact cell viability during the experimental set-up (*Figure 3—figure supplement 1* and *Figure 3—videos 1 and 2*). We monitored control and NZ treated cells for 16 hr using our previously described feedback routine to assess their migration properties. As seen from the corresponding morphodynamic maps, NZ treated cells still show protruding activity that, however, is less sustained and connected giving rise to separated patches of activity (*Figure 3A–B* and *Figure 3—figure supplements 2 and 3*). We measured the average protrusion speed for each single cell with or without treatment (*Figure 3C*) and found no significant difference, showing that the protrusive ability of NZ treated cells was not altered. As a consequence, the average instantaneous speeds of cells in a short 5 min time window are the same in both conditions (*Figure 3D*). Yet, the directionality ratio – defined as the cell displacement divided by the length of the cell trajectory – strongly differs, pointing to a difference in the directionality of movement (*Figure 3E*). Thus, NZ treated cells are less persistent than control cells, as directly observed from their trajectories (*Figure 3F–G* and *Figure 3—video 1*). We further quantified the persistence of migration by measuring the autocorrelation of direction of movement, which takes into account only the angle of direction of a moving cell and correlates it over time (*Gorelik and Gautreau, 2014*). The decay of this autocorrelation informs on the timescale over which cells randomize their direction of movement (*Figure 3H*). By fitting an exponential function on the autocorrelation curves of single cells, we extracted a characteristic persistence time of each cell (*Figure 3I*). Control cells are persistent over ~2.5 hr on average, whereas NZ treated cells are persistent over 20 min on average only. Taken together, our results showed that NZ treated cells are protruding as efficiently as control cells, but in a nonco-ordinated manner over the timescale of more than 20 min. It thus suggests that cell internal organization is required for long-term coordinated protruding activity and persistent cell migration.

## Persistent cells show sustained polarized trafficking from the Golgi complex to protrusions

The Golgi complex plays important roles in directed secretion of vesicles and cargos to the leading edge of migrating cells (*Yadav et al., 2009*). To assess the dynamics of Golgi-derived secretion, we followed synchronized secretion of collagen X from the ER to the plasma membrane using the Retention Using Selective Hooks (RUSH) assay. To better visualize the sites of collagen X arrival we combined RUSH with selective protein immobilization (SPI) on the coverslip via antibody capturing

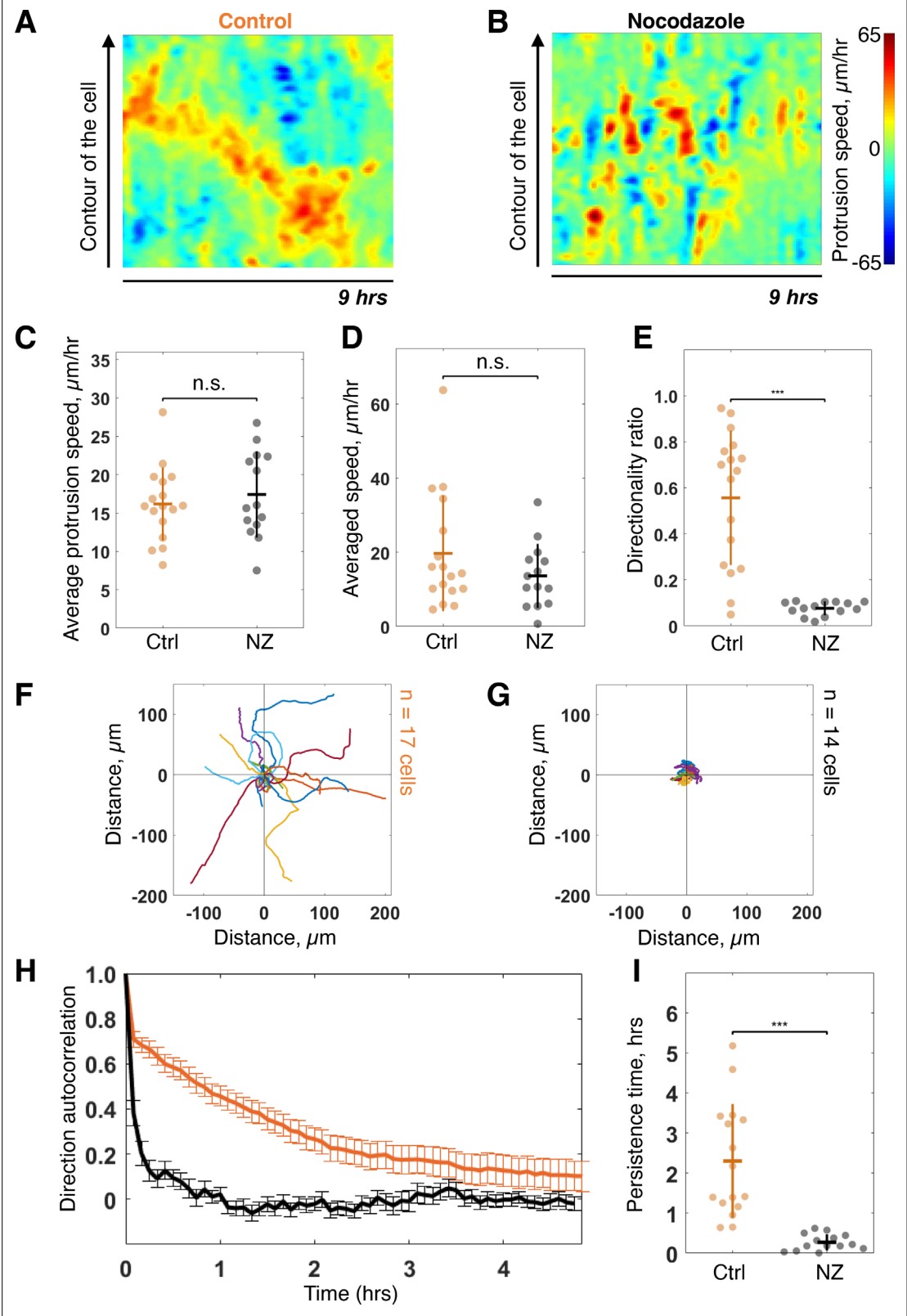

**Figure 3.** Low dose of Nocodazole (NZ) reduces persistence of migration. (**A–B**) Representative morphodynamic maps of RPE1 cells freely moving on a fibronectin-covered coverslip in control condition (Ctrl) (**A**) and with NZ (0.1 μm) (**B**). (**C–E**) Average protrusion speed (**C**), average cell speed (**D**), and directionality ratio (**E**) compared in Ctrl and with NZ (Wilcoxon rank sum test, *p≤0.05, **p≤0.01, ***p≤0.001). (**F–G**) Trajectories of RPE1 cells in Ctrl (n =

*Figure 3 continued on next page*

*Figure 3 continued*

17) (**F**) and with NZ (n = 14) (**G**) (trajectories plotted over 7 hr of experiment). (**H–I**) Direction autocorrelation (**H**), and persistence time (**I**) compared in Ctrl and with NZ (Wilcoxon rank sum test, *p≤0.05, **p≤0.01, ***p≤0.001). Data used for **C-I** and related scripts can be found in *Figure 3—source data 1*.

The online version of this article includes the following video, source data, and figure supplement(s) for figure 3:

**Source data 1.** Data and analysis scripts with explanations for *Figure 3* and its supplements.

**Figure supplement 1.** Low concentrations of Nocodazole (NZ) slow down microtubule (MT) dynamics without significantly changing the MT network content.

**Figure supplement 2.** All morphodynamic maps from control experiments.

**Figure supplement 3.** All morphodynamic maps from Nocodazole experiments.

**Figure 3—video 1.** RPE1 cell freely moving in 2D in control conditions and with Nocodazole (0.1 μM).
https://elifesciences.org/articles/69229/figures#fig3video1

**Figure 3—video 2.** Microtubule dynamics in RPE1 cells in control conditions and with Nocodazole (0.1 μM).
https://elifesciences.org/articles/69229/figures#fig3video2

that shows secretion of collagen X at ~20 min after release from the ER (see Materials and methods) (*Fourriere et al., 2019*). Interestingly, although we could detect secreted cargos accumulating in the direction of the Nucleus-Golgi axis (in 6 out of 52 cells), (*Figure 4—figure supplement 1A* for representative examples), in 20 out of 52 cells, the accumulated cargo did not align with the Nucleus-Golgi axis, probably because we captured cells while turning (*Figure 4A* at 42 min when the cargo accumulates in the Golgi). Importantly, we observed the accumulation of secreted cargos toward the newly formed protrusions in all these cases (6 + 20 out of 52 cells in total) (*Figure 4A* at 1 hr, *Figure 4—figure supplement 1A* and *Figure 4—video 1*). To further follow constitutive Golgi-derived trafficking activity at a longer timescale, we used Rab6 as general marker for Golgi-derived secretion (*Fourriere et al., 2019*). We engineered a CRISPR knock-in cell line with an iRFP fluorescent protein fused to the endogenous Rab6A protein (see Materials and methods). Compared to these CRISPR knock-in cells (and wild-type [WT] hTERT-RPE1 cells), GFP-Rab6 overexpressing cells were more persistent (*Figure 4—figure supplement 3*), likely due to increased secretory activity in overexpressing cells. Using highly inclined and laminated optical sheet (HILO) microscopy, we could minimize the signal from the Golgi and enhance signal from the Rab6 vesicles (*Figure 4B*, **top** and *Figure 4—video 1*). Moreover, we further suppressed the signal from the Golgi complex by segmenting and masking it to quantify only cell trafficking. We performed the morphodynamic map analysis of cell protrusions (*Figure 4C*, **top**), in addition to a Rab6- trafficking map (*Figure 4C*, **bottom**). The latter was computed by measuring the average intensity along lines from the Golgi centroid to the cell contour (*Figure 4B*, **bottom**). The color code from blue (no Rab6-signal) to red (max Rab6-signal) of this trafficking map shows the hotspots of trafficking as a function of time. We found that the morphodynamic and trafficking maps correlate (*Figure 4D*) confirming a sustained trafficking to protrusions at the long timescale. By performing a temporal cross-correlation analysis, we observed a peak that occurs at a positive time lag of 19 ± 11 min indicating that protrusions precede trafficking. This delay is also obvious from the alignment of the morpho and trafficking maps when a cell reorients (e.g. *Figure 4C*, **black dashed lines**). To further test how internal organization impacts polarized trafficking, we analyzed the secretion of collagen X and the flow of Rab6-positive vesicles in NZ treated cells. As expected, in these cells there was no polarized secretion of vesicles (n = 18, see *Figure 4—figure supplement 1B* and *Figure 4—video 1* for representative examples) and no polarized trafficking of vesicles, but an isotropic directed flow toward the membrane (n = 14 cells, see *Figure 4—video 1* for a representative example). To further investigate the role of Golgi-based trafficking, we used Golgicide A, which perturbs secretion from the Golgi by specifically targeting GBF1, a GEF of Arf1 for the COPI coat production at the Golgi. We found that treatment with Golgicide A reduces persistence (*Figure 4—figure supplement 4*), however, less than treatment with either NZ or Taxol, a MT stabilizing drug that interferes with MT dynamics. Along the same line, as mentioned above, overexpression of Rab6 leads to increase in persistence of migration (*Figure 4—figure supplement 3*).

Together, these results show that intracellular polarity axis is required to keep trafficking aligned with protrusive activity over timescales longer than ~20 min. Moreover, we found that interfering with MT dynamics had a stronger effect on persistent migration than interfering with secretion activity from the Golgi.

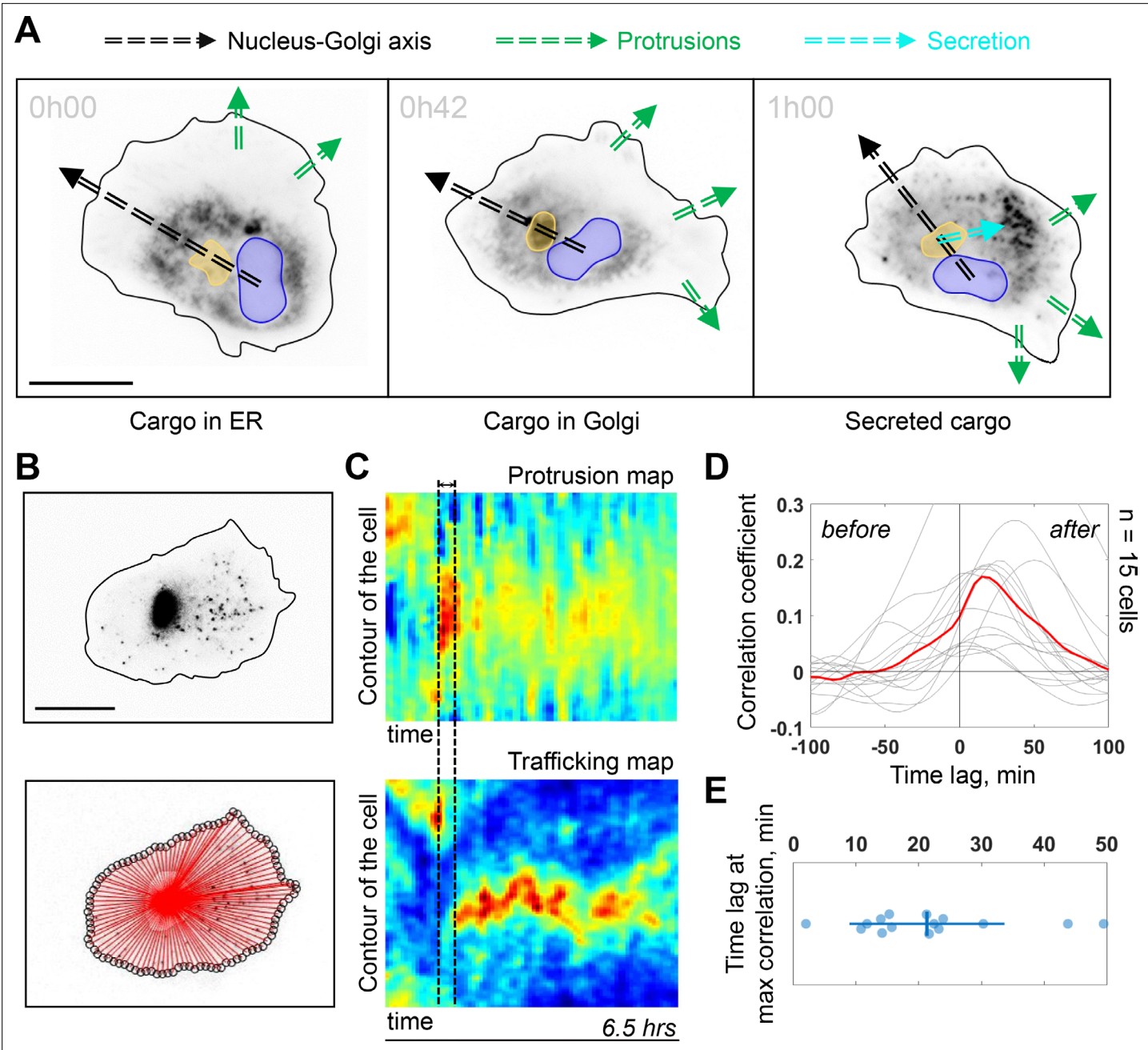

**Figure 4.** Trafficking from Golgi complex is biased toward the protrusion. (**A**) Collagen X cargo (labeled in black) is traveling from ER to the Golgi complex and secreted during a Retention Using Selective Hooks assay experiment (blue contour: nucleus, yellow contour: Golgi complex, green dashed line: protrusions, black dashed line: Nucleus-Golgi axis, cyan dashed line: secretion axis, scale bar - 20 μm). (**B**) Top, RPE1 cells expressing endogenous levels of a marker for post-Golgi vesicles (iRFP-Rab6A) (scale bar - 20 μm). Bottom, red lines represent the lines over which vesicle traffic intensity is calculated over time. (**C**) Top, a representative morphodynamic map representing plasma membrane protrusions in time. Bottom, a representative trafficking map showing the flow of post-Golgi vesicles from the Golgi complex along straight lines toward the surface over time time period - 6.5 hr, black dashed lines represent the time difference between a spike in protrusions (top) and a spike in secretion (bottom). X-axis represents time and Y-axis represents the contour of the cell. (**D**) Cross-correlation coefficient between plasma membrane protrusions and secretion as a function of the time lag (n = 15 cells, red line: average curve depicting the correlation coefficient, gray lines: single cell data; 'before' and 'after' denote the time before the protrusion peak and after, respectively). (**E**) Single cell time lags in minutes between protrusions and secretion at maximal correlation, obtained by fitting the peak of individual cross-correlation curves (gray curves in (**D**)). Data used for **C-E** and related scripts can be found in *Figure 4—source data 1*.

The online version of this article includes the following video, source data, and figure supplement(s) for figure 4:

**Source data 1.** Data and analysis scripts with explanations for *Figure 4* and its supplements.

*Figure 4 continued*

**Figure supplement 1.** Examples of Retention Using Selective Hooks-SPI assays in control and Nocodazole conditions.

**Figure supplement 2.** All protrusion and trafficking maps from trafficking experiments in control conditions.

**Figure supplement 3.** Comparison of persistence between Rab6 overexpression and Rab6 endogenous level.

**Figure supplement 4.** Large scale comparison of persistence for different drug conditions.

**Figure 4—video 1.** RPE1 cells in Retention Using Selective Hooks (RUSH)-SPI assay with labeled Collagen X cargo in control and with Nocodazole, and with labeled Rab6A in control and with Nocodazole.

https://elifesciences.org/articles/69229/figures#fig4video1

## An imposed Cdc42 gradient reorients the Golgi complex

Because protrusion activity preceded trafficking from the Golgi, we next investigated how protruding activity regulates internal polarity. We controlled the protrusive activity with optogenetic stimulation while monitoring internal polarity. For the optogenetics, we used the iLID/SspB dimerizing system (*Guntas et al., 2015*) to locally activate Cdc42 (*Valon et al., 2015*) by recruiting the catalytic DH-PH domain of ITSN (Intersectin) – one of its specific activators – using localized blue light illumination. We used the Nucleus-Golgi axis as a proxy of the internal polarity axis. Using the previously described feedback imaging routine, we added a possibility to induce Cdc42 activation while imaging a migrating cell and adapt the activation pattern to the changing shape of the cell (see *Figure 1—figure supplement 1B* and Materials and methods). Our previous experiments revealed that a sharp gradient of Cdc42 is the most effective to control directionality of cell movement (*de Beco et al., 2018*), therefore, we chose to activate a thin region along the border of the cell. We conducted 16 hr live cell imaging experiments, starting with the activation 90° away from the existing Nucleus-Golgi axis. We found that optogenetic activation of Cdc42 was sufficient to reorient the Nucleus-Golgi axis toward the region of activation (*Figure 5A* and *Figure 5—video 1*). We quantified the rotation of the Nucleus-Golgi axis toward the axis of the optogenetic activation (going from the center of nucleus to the center of activation area) over time for 19 cells and observed a systematic reorientation in 3 hr followed by a stabilization of the Nucleus-Golgi axis around 0° (18° ± 28°, after 4 hr, *Figure 5B* and *Figure 5—figure supplement 1C*). To test the specificity of the Cdc42 activation, we performed control experiments (n = 26 cells), in which the DH-PH domain of ITSN is missing (see scheme in *Figure 5C*, **top**). In control experiments, the Nucleus-Golgi axis constantly moved without stabilization at the axis of optogenetic activation (0°) (33° ± 172°, after 4 hr, *Figure 5C*). To further confirm that the optogenetic activation stabilized the Nucleus-Golgi axis, we optogenetically activated cells in front of existing Nucleus-Golgi axis (*Figure 5—figure supplement 1A*). The axis was stabilized as the angle between Nucleus-Golgi axis and optogenetic activation stayed close to 0° during the full duration of the experiment (n = 19 cells) (8° ± 34°, after 4 hr, *Figure 5—figure supplement 1B*). Since our optogenetic activation leads to protrusions and cell migration, the Nucleus-Golgi axis may reorient in a passive manner through cell shape changes. To better control cell shape, we performed similar experiments on round fibronectin micropatterns. Similar to nonpatterned cells, the reorientation of Nucleus-Golgi axis aligned with the activation area (*Figure 5—figure supplement 1D-F*). Yet, we found that the Nucleus-Golgi axis reorientation happened faster, with 50% of cells reorienting in 1 hr on a pattern compared to 3 hr when freely moving (*Figure 5—figure supplement 1C*). Thus, our results show that a biochemical Cdc42 activity but not a change in cell shape is able to reorient the Nucleus-Golgi axis toward it and then stabilize it.

## An imposed Cdc42 activation rescues persistent cell migration

We found that persistent migration requires stable protrusive activity, which is lost upon NZ treatment. We thus tested if we could rescue this loss of protrusive stability using activation of Cdc42 by optogenetics. We used optogenetic activation of Cdc42 in presence of NZ (0.1 μM) and found that optogenetic Cdc42 activation indeed rescued the persistence of cell migration, as observed directly from cell trajectories (*Figure 5D*) or from the directionality ratios, which are calculated by taking the ratio of the displacement of the cell and the length of the actual path it took (*Figure 5E*). Whereas directionality ratio in presence of NZ was drastically perturbed, reaching only 19% of directionality in control cells (directionality ratio of 0.11 ± 0.04), optogenetic Cdc42 activation restored the directionality to 0.4 ± 0.2, a number comparable to freely migrating cells (0.59 ± 0.31) and similar to optogenetically

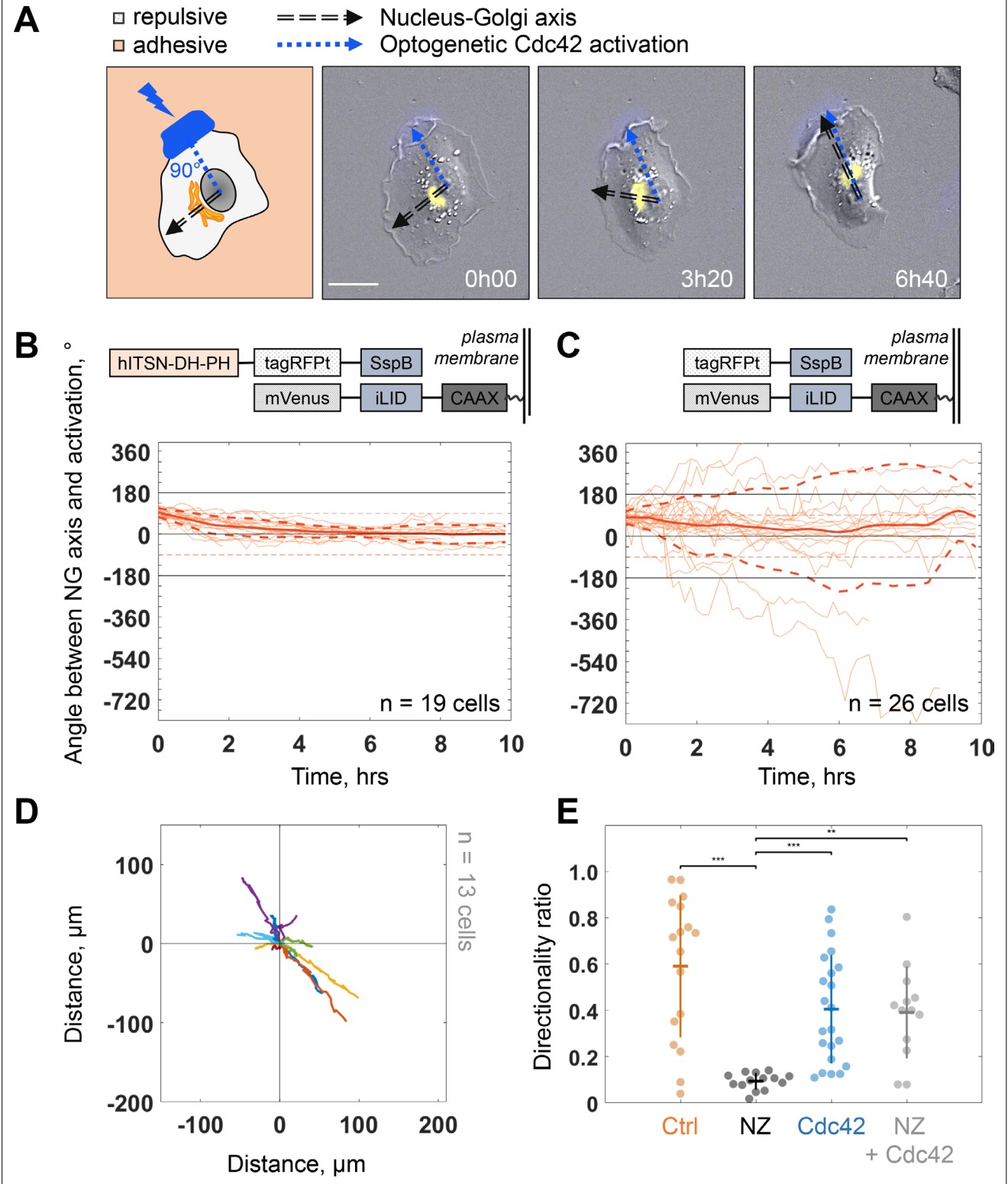

**Figure 5.** Biochemical gradient of Cdc42 reorients the Golgi complex and rescues directional migration. (**A**) DIC image overlaid with Golgi marker (yellow, iRFP-Rab6A) of an RPE1 cell optogenetically activated 90° away from its initial Nucleus-Golgi axis (black dashed line: Nucleus-Golgi axis, blue dashed line: optogenetic activation axis, scale bar – 20 µm). (**B–C**) Optogenetic activation of Cdc42 90° away from Nucleus-Golgi axis leads to its reorientation in RPE1 cells freely moving on fibronectin covered coverslip (n = 19 cells) (**B**) and is random in control condition (n = 26 cells) (**C**) (thin orange lines: single cell data, thick orange line: data average, dashed thick orange lines: standard deviation; corresponding optogenetic constructs

*Figure 5 continued on next page*

*Figure 5 continued*

used are depicted above the graphs). (**D**) Trajectories of cells moving in this experimental condition (n = 13 cells; trajectories plotted over 7 hr of experiment). (**E**) Directionality ratio comparison between optogenetically activated cells in presence of Nocodazole (NZ) (orange: freely moving cells ('Ctrl', n = 17 cells), black: freely moving cells in presence of NZ ('NZ', n = 14 cells), blue: optogenetically activated cells ('Cdc42', n = 22 cells), gray: optogenetically activated cells in presence of NZ ('NZ +Cdc42, n = 13 cells); Kruskal–Wallis test followed by a post hoc Dunn's multiple comparison test, *p≤0.05, **p≤0.01, ***p≤0.001). Data used for **B-E** and related scripts can be found in *Figure 5—source data 1*.

The online version of this article includes the following video, source data, and figure supplement(s) for figure 5:

**Source data 1.** Data and analysis scripts with explanations for *Figure 5* and its supplements.

**Figure supplement 1.** Biochemical gradient of Cdc42 stabilizes the Golgi complex position and reorients it on an isotropic pattern.

**Figure 5—video 1.** RPE1 cells exposed to local optogenetic Cdc42 activation while freely moving, on a round pattern and freely moving with Nocodazole.

https://elifesciences.org/articles/69229/figures#fig5video1

activated cells without NZ (0.4 ± 0.24) (*Figure 5E*). The fact that we could rescue persistent migration in NZ treated cells indicates that the loss of persistency in NZ treated cells is the consequence of the absence of a mechanism stabilizing the protrusive activity and not an inherent inability of cells to move persistently.

## A minimal model coupling protrusive activity and polarized trafficking recapitulates persistent migration

Our results indicate that persistent mesenchymal migration emerges from a feedback between the alignment of the internal polarity axis by Cdc42 and stabilization of Cdc42-dependent protruding activity through polarized trafficking toward protrusions (*Figure 6A*). We constructed a minimal physical model to know whether this feedback is enough by itself to recapitulate the features we observed with the persistently migrating RPE1 cells (see *Figure 6—figure supplement 1* and Materials and methods for a detailed explanation of the model). To implement the two sides of the feedback with minimal settings, we chose to model synthetic morphodynamic maps that advantageously capture quantitatively the process of cell migration in a single piece of data.

We first implemented the morphodynamic map corresponding to a single event of protrusive activity, which may comprise several protrusion/retraction cycles. Membrane dynamics following a pulse activation of Cdc42 and Rac1 were previously experimentally obtained and described in *Yamao et al., 2015*. In this work, the authors computed the transfer function between a point-like RhoGTPase activity at time 0 and position 0, and the membrane dynamics that follow. We numerically synthesized this transfer function and a Cdc42 pulse of activity extended in space and time (*Figure 6B*) such that the convolution of the two leads to a morphodynamic map similar to a single event of protrusive activity as seen in our data (*Figure 6C*). Next, we simulated full morphodynamic maps by nucleating protrusive events randomly in space and time such that the frequency of protrusive activity of our model matched the data (*Figure 6D*). On these maps, we assumed that the cell possessed an internal polarity axis parametrized by a moving point, $c_{axis}(t)$, on the y-axis, which corresponds to the intersection between the polarity axis and the cell contour. For sake of simplicity, we did not make the distinction between the axis of directed trafficking/secretion and Nucleus-Golgi axis but considered a single effective one. We then implemented the feedback between polarity axis and protrusion dynamics. For the first side of the feedback, we introduced a probability $P_{polarized}$ to nucleate a protrusive event in front of the polarity axis and a probability $P_{rand} = 1 - P_{polarized}$ elsewhere. When $P_{polarized} = 0$, protrusions are happening randomly along the contour, and when $P_{polarized} = 1$, protrusions are always happening in front of the polarity axis (*Figure 6E*). For the second side of the feedback, we assumed that the polarity axis was pulled toward the protrusion by an effective force $F_{prot}$ that acts against a force $F_{basal}$ characterizing the random rotation of the polarity axis. The bias of the protrusive activity on the polarity axis positioning can be parametrized by a number $\kappa$ such that $\frac{d}{dt}c_{axis}(t) = \kappa F_{prot} + (1 - \kappa) F_{basal}$. When $\kappa = 0$, the polarity axis follows its natural evolution, and when $\kappa = 1$, the polarity axis follows the protrusive activity (*Figure 6F*). The strength of both sides of the feedback can thus be summarized by two numbers between 0 and 1. For a given value of these two numbers, we could simulate realistic morphodynamic maps ranging from nonpersistent to persistent migrating cells (*Figure 6G* and *Figure 6—video 1*). From these maps, we generated cell

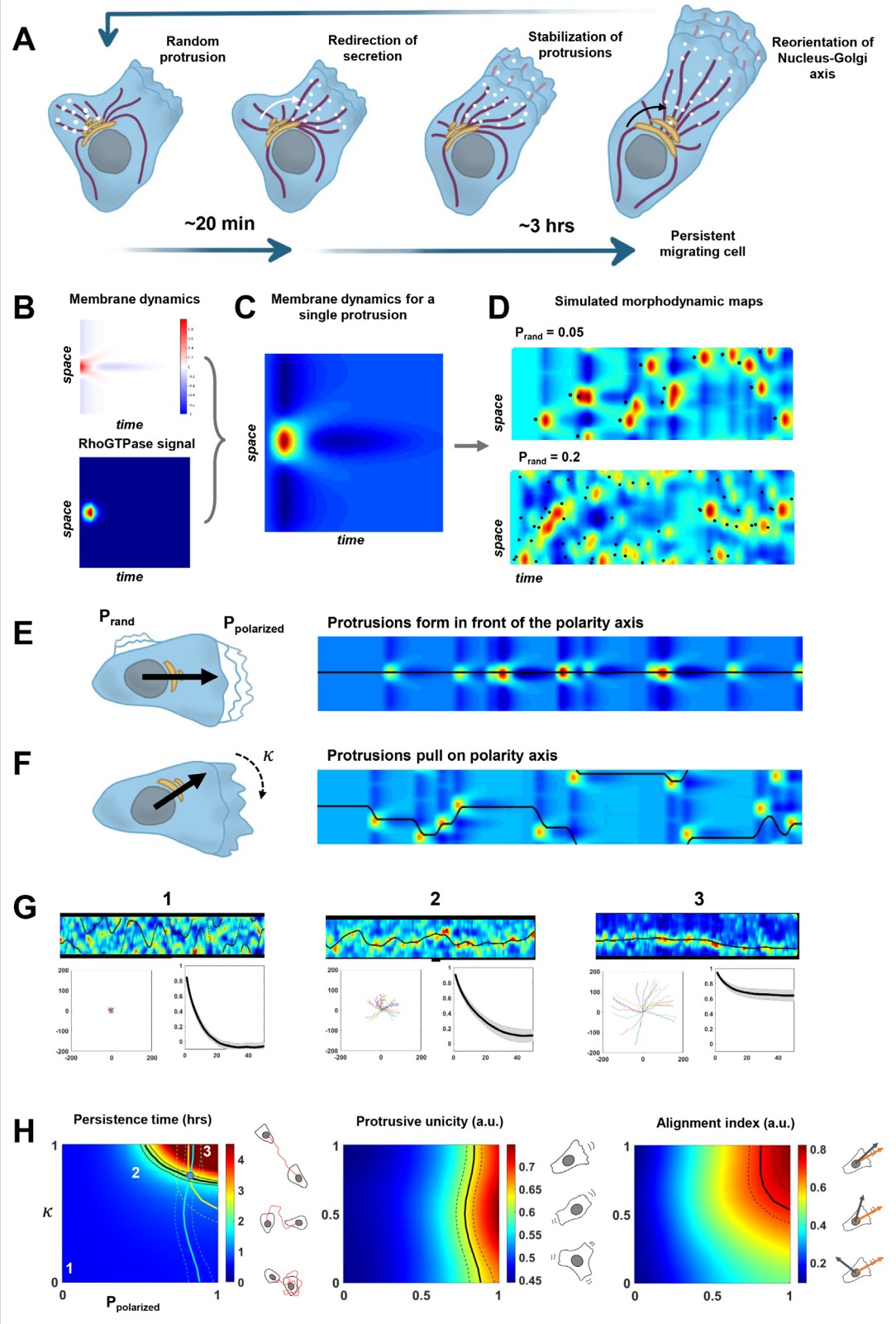

**Figure 6.** A minimal physical model based on the coupling between protrusive activity and internal polarity recapitulates persistent migration. (**A**) Scheme representing the sequence of events leading to persistent migration. The Golgi complex is in yellow, microtubules are in violet, and vesicles in white. (**B–C**) Membrane dynamics for a point-like Cdc42 activation (A, upper panel) were convolved with a RhoGTPases signal (A, lower panel) to compute membrane dynamics for a single protrusive event (**B**). (**D**) Overall synthetic morphodynamic maps generated by varying the protrusive activity

*Figure 6 continued on next page*

*Figure 6 continued*

frequency. The intensity of protrusive activity varies from one protrusion every 20 frames (top), to one every 5 frames (bottom), the latter value having been retained for our simulations in (**G–H**). (**E–F**) Implementation of the feedback, with $P_{polarized}$ quantifying the probability to form a protrusion in front of the polarity axis (equal to one in the simulated morphodynamic map in (**E**)), and $\kappa$ quantifying the capacity of protrusions to pull on the polarity axis (equal to one the simulated morphodynamic map in (**F**)). (**G**) Examples of morphodynamic maps (top, black line is $c_{axis}(t)$ the direction of the internal polarity axis), cell trajectories (bottom left), and autocorrelation of direction (bottom right) for different values of strength of the feedback. 1: $P_{polarized} = 0$; $\kappa = 0$. 2: $P_{polarized} = 0.5$; $\kappa = 0.8$. 3: $P_{polarized} = 0.8$; $\kappa = 1$. (**H**) Phase diagrams of persistence time, protrusive unicity, and alignment index. Black lines in the diagrams correspond to the experimental values (persistent time ~2 hr, protrusive unicity ~0.6 and alignment index ~0.7). The lines of the two last diagrams are reported on the first one, where they cross at a single point (blue dot), hereby showing consistency between the model and our experimental measurements. Data used for **A-H**, related scripts and additional explanations can be found in *Figure 6—source data 1*.

The online version of this article includes the following video, source data, and figure supplement(s) for figure 6:

**Source data 1.** Data and analysis scripts with explanations for *Figure 6* and its supplements.

**Figure supplement 1.** Detailed explanation of the minimal physical model.

**Figure supplement 2.** Comparison of phase diagrams of HeLa and RPE1 cells.

**Figure supplement 3.** Comparison of phase diagrams of Nocodazole and Golgicide A treated RPE1 cells.

**Figure 6—video 1.** Movement of a synthetic cell recapitulated by the minimal physical model.

https://elifesciences.org/articles/69229/figures#fig6video1

trajectories from which we computed the autocorrelation of direction and persistence time, as for our experimental data (*Figure 6G*). In addition, we computed two other independent parameters aimed at quantifying cell polarity (*Figure 6H*). The protrusive unicity index characterizes how many distinct protrusive activities are competing at a given time, and the alignment index characterizes how well the polarity axis aligns with the direction of movement.

Running our simulation for all possible values of $P_{polarized}$ and $\kappa$, we obtained three phase diagrams for the persistence time, protrusive unicity, and alignment index (*Figure 6H*). These 'look-up' tables differ in their dependencies with regard to the two parameters, and can, thus, be used to estimate their values independently. When combined, they should converge to a single couple of values. If it is the case, it would be a signature of the consistency of our minimal model. It was indeed the case for our data on RPE1 cells, where we found a persistence time of 2.3 ± 1.4 hr, a protrusion unicity index of 0.65 ± 0.15, and an alignment index of 0.72 ± 0.21. Using these numbers and the phase diagrams to estimate $P_{polarized}$ and $\kappa$, we obtained a region of the parameter space that is consistent and predicts that $\kappa = 0.9$ and $P_{polarized} = 0.7$. Thus, our model suggests that the high persistence of RPE1 cells can be explained by a relatively high values of the feedback strengths. Next, we used the same approach on HeLa cells, a cell line that is less persistent than RPE1 (*Figure 6—figure supplement 2*). Our model indicates that the decreased persistency of HeLa cells can be explained by a lower value of $P_{polarized} = 0.35$, reflecting the multiple competing ruffling fronts observed in these cells. To further exploit the quantitative aspect of our model, we also analyzed our NZ and Golgicide A datasets (*Figure 6—figure supplement 3*). For those conditions, unfortunately, the alignment index cannot be calculated, because the Golgi complex is dispersed. Yet, from the persistence time and protrusive unicity, the model predicts that NZ has a stronger effect on $P_{polarized}$ and $\kappa$ than Golgicide A, confirming the central role of MTs in the feedback.

## Discussion

In this work, we showed that persistent mesenchymal migration observed on a timescale of several hours can emerge from a feedback between protrusion dynamics and polarized trafficking. This feedback mechanism corresponds to the one that has been intensively documented in yeast, where polarized bud formation is dynamically maintained by coupling of transport and signaling (*Eugenio et al., 2008*). Using experimental approaches, we showed that protrusion dynamics and polarized trafficking are coupled in mesenchymal RPE1 cells. We demonstrated using optogenetics that sustained local activation of Cdc42 is sufficient to reorient the Nucleus-Golgi axis in 3 hr (*Figure 5*). Moreover, we showed that the Nucleus-Golgi axis correlates with the direction of migration and the trafficking of Rab6 secretory vesicles in freely migrating cells (*Figures 1 and 4*) and that protrusions precede trafficking. Together, our results suggest that a sustained protrusive activity reorients the trafficking and secretory pathway toward protrusions. We observed a time lag of 20 min between protrusions and

redirection of the trafficking of Rab6-positive vesicles and that secretion was preferentially directed to newly formed protrusions. Note that whereas Rab6 marks the specific trafficking from the Golgi complex, the Nucleus-Golgi polarity axis could additionally represent other polarized trafficking, independent of the Golgi complex. Indeed, the recycling compartment regulated by Rab11 also aligns with the Nucleus-Golgi polarity axis of the cell and could additionally contribute to polarized trafficking (*Ferro et al., 2021*). Taken together, these results strongly support the fact that protrusions orient polarized trafficking on a short timescale, and orient the Nucleus-Golgi axis on a longer one.

The Rho GTPase Cdc42 was already proposed to play a role in the Nucleus-Golgi axis reorientation (*Etienne-Manneville and Hall, 2001*), and several different pathways activated by Cdc42 could be working in the process. MTs, that are anchored to the protrusion by forming focal adhesions, have been proposed to pull toward the protrusion hereby reorienting the centrosome and the Golgi complex together (*Etienne-Manneville et al., 2005*). The Golgi complex could be pulled by polymerizing actin forces via GOLPH3/MYO18A pathway (*Xing et al., 2016*). Eventually, the actin retrograde flow could push the nucleus backward (*Gomes et al., 2005*). Our cargo trafficking experiments have indicated that the cargo is trafficked and secreted toward the newly forming nascent protrusions. This could be explained by the MTs being guided toward the newly forming adhesions by the actin cytoskeleton (*Etienne-Manneville, 2013*; *Meiring et al., 2020*). Fourriere et al. has also shown that Rab6-positive post-Golgi vesicles release their cargo in the vicinity of FAs (Focal Adhesions) (*Fourriere et al., 2019*). Further investigation is needed to reveal the exact mechanisms by which Rab6-positive vesicles accumulate at membrane ruffles forming the protruding front of the cell, before the Nucleus-Golgi axis reorients. One hypothesis could be that the MT density is higher on the side of the protruding front (*Etienne-Manneville, 2013*; *Meiring et al., 2020*). Alternatively, post-translational modification of a subset of MTs via acetylation of α-tubulin, which has been found to accumulate in cell protrusions and to regulate cell polarization (*Montagnac et al., 2013*), could lead to preferential trafficking of Rab6-positive vesicles. It has been proposed that Rab6-positive vesicles fission from Golgi/TGN at a limited number of hotspot sites, to regulate their exit along MTs (*Miserey-Lenkei et al., 2017*). Our data, showing that reorganization of secretion precedes the reorientation of the Golgi complex, is consistent with the fact that the MTs, which direct secretion, reorganize before the Golgi complex is reoriented.

On the other hand, our data demonstrate that polarized trafficking sustained protruding activity: using low doses of NZ to disrupt internal cell organization and polarized trafficking (*Figure 4—figure supplement 1B* and *Figure 4—video 1*), we showed that the persistence time of cell migration dropped from 2.3 hr to 20 min (*Figure 3*). Interestingly, protrusion speed and instantaneous cell speed were not affected, showing that the loss of persistency was not due to cell's inability to protrude or move. Our rescue experiment using constant Cdc42 activation (*Figure 5*) confirmed that sustaining protruding activity is sufficient for persistent migration. Of note, NZ treated cells were still persistent over 20 min, showing that in this condition the protrusive activity is stable for a longer time than the duration of a single protrusion-retraction event (on the order of 100 s). This also suggested that RhoGTPases' activities are comparable between WT and mildly NZ treated cells, although we cannot exclude that endogenous associated GEF/GAP activators and deactivators were affected. Thus, NZ treated cells are still able to stabilize a protrusive activity over several cycles, possibly thanks to the existence of a vimentin template (*Gan et al., 2016*). However, NZ treated cells were not able to stabilize their protrusive activity over longer time ( >20 min), which we attributed to the loss of polarized trafficking (*Figure 4—video 1*), potentially of protrusion-promoting factors toward the cell front in Rab6-positive vesicles (*Figure 4*). Rab6 has been proposed to be a general regulator of post-Golgi secretion, and it has been shown that irrespective of the transported cargos most Rab6-positive carriers are not secreted randomly at the cell surface, but on localized hotspots juxtaposed to focal adhesions (*Fourriere et al., 2019*). However, we cannot exclude that protrusion-promoting factors are transported from other compartments, such as the recycling endosomal compartment that is found at the proximity of the Golgi complex, and, thus, also polarizes along the Nucleus-Golgi axis.

We recapitulated the two sides of the feedback in the framework of a new minimal physical model. This model is based on the coupling between an internal polarity axis – a vector, and protrusion dynamics modeled by synthetic morphodynamic maps. This model can be thought of as a 'cell compass', where protrusions pull on the needle that has some inertia, and the direction of the needle locally promotes the initiation of protrusions. Even if we do not specify the exact nature of

the polarity axis in the model, our data suggest that the relevant axis is the direction of secretion, which follows protrusions with a 20 min delay. The Nucleus-Golgi axis also follows the direction of motion powered by protrusions, albeit with a longer delay, but neither the Nucleus-Golgi axis, nor the direction of secretion appears to be instructive. Yet, for stretches of persistent motion when the direction of secretion, Nucleus-Golgi axis, and direction of migration are all aligned, the Nucleus-Golgi axis is a good proxy of the polarity axis. Many mathematical models of cell polarization (*Jilkine and Edelstein-Keshet, 2011*; *Mogilner et al., 2012*) or of cell migration (*Danuser et al., 2013*) were previously introduced, but our approach differs in the sense that the whole cell migratory behavior can be described by only two effective parameters that quantify the strength of the feedback. We showed that the persistence time of different cell types can be predicted from the measurement of two parameters – the average number of competing protrusions and the alignment of polarity axis with direction of motion (*Figure 6*). Interestingly, RPE1 cells sit in the persistence time phase diagram at the relatively sharp transition between nonpersistency and superpersistency. This suggests that cells might be tuned at an optimal functioning point, to be persistent in their migration while not being locked into a straight path, possibly to be able to respond to environmental cues. One way to test this prediction would be to experimentally modify the strengths of the feedback: a slight increase of $\kappa$ and $P_{polarized}$ should lead to superpersisters. This could be achieved by expressing in cells a fusion between ITSN-DHPH domain, an activator of Cdc42, and Rab6 or Rab11 to reinforce the feedback loop between protrusions and secretion. Our model shows that cells can polarize with a unique protruding front (protrusive unicity close to 1) when the probability to form a protrusion in front of the polarity axis is high ($P_{polarized} = 1$). Of course, this was expected since we assumed that there was a unique polarity axis. In our experiments, we indeed observed a unique axis of polarized trafficking (*Figure 4*), and it remains to be understood how cells achieve this unicity. As a matter of fact, we could imagine that multiple protrusions would be sustained simultaneously, associated to their own polarized trafficking routes, and with the same feedback mechanism being involved. A possible answer would be the existence of a limiting component in the system, as proposed in the context of yeast polarity (*Chiou et al., 2017*). Alternatively, the level of RhoGTPase activity might be tuned to limit the number of competing protrusions, as suggested by the relationship between Rac1 activity and directional persistent migration (*Pankov et al., 2005*).

To conclude, our present work focused on the coupling between protrusion dynamics and polarized trafficking. Many other functional units supporting cell polarity are likely to be involved (*Vaidžiulytė et al., 2019*), and it will be interesting to see in future studies how other coupling mechanisms can contribute to the robustness of cell polarity during persistent migration. Additionally, it would be of interest to see how our conclusions can be extended to other types of migration, such as the amoeboid one.

# Materials and methods

**Key resources table**

| Reagent type (species) or resource | Designation | Source or reference | Identifiers | Additional information |
|---|---|---|---|---|
| Cell line (*Homo sapiens*) | hTERT RPE1 (immortalized, normal, female) | ATCC | ATCC Cat# CRL-4000, RRID:CVCL_4388 | |
| Cell line (*H. sapiens*) | HeLa (adenocarcinoma, female) | ATCC | ATCC Cat# CCL-2, RRID:CVCL_0030 | |
| Cell line (*H. sapiens*) | RPE1::iRFP-Rab6A | This paper | | Heterozygous iRFP-Rab6A CRISPR knock-in |
| Cell line (*H. sapiens*) | RPE1::iRFP-Rab6::hITSN1-tgRFP-SSPB wtt::Venus-iLID-CAAX | This paper | | Stable cell line with heterozygous iRFP-Rab6A label and lentivirally induced expression of optogenetic constructs |
| Cell line (*H. sapiens*) | RPE1::GFP-Rab6A | This paper | | Lentivirally induced stable GFP-Rab6A overexpression |

*Continued on next page*

*Continued*

| Reagent type (species) or resource | Designation | Source or reference | Identifiers | Additional information |
|---|---|---|---|---|
| Cell line (*H. sapiens*) | RPE1::GFP-Rab6A::myr-iRFP | This paper | | Stable cell line with GFP-Rab6A and myr-iRFP overexpression |
| Cell line (*H. sapiens*) | RPE1::EGFP-α-tubulin | Piel lab | | Stable cell line with a α-tubulin marker |
| Cell line (*H. sapiens*) | RPE1::EB3-EGFP | https://doi.org/10.1038/nmeth.1493 | | Krek lab |
| Transfected construct (*H. sapiens*) | pLL7.0: hITSN1(1159–1509)-tgRFPt-SSPB WT | Addgene | RRID: Addgene_60419 | Lentiviral optogenetic construct for Cdc42 activation |
| Transfected construct (*H. sapiens*) | pLL7.0: Venus-iLID-CAAX | Addgene | RRID: Addgene_60411 | Lentiviral optogenetic construct |
| Transfected construct (*H. sapiens*) | pMD2.G | Addgene | RRID: Addgene_12259 | Lentiviral VSV-G envelope expressing plasmid |
| Transfected construct (*H. sapiens*) | psPAX2 | Addgene | RRID: Addgene_12260 | Second generation lentiviral packaging plasmid |
| Transfected construct (*H. sapiens*) | pHR: myr-iRFP | Coppey lab | | Lentiviral construct to label plasma membrane with iRFP |
| Transfected construct (*H. sapiens*) | pEGFP-C3: Rab6A_wt | Goud lab | | Plasmid construct to label Rab6A (Golgi complex) |
| Transfected construct (*H. sapiens*) | pIRESneo3: Str-KDEL-SBP-EGFP-COL10A1 | https://doi.org/10.1083/jcb.201805002 | | RUSH system plasmid with EGFP tagged Collagen X cargo |
| Antibody | anti-GFP (rabbit monoclonal) | Recombinant Antibody Platform of Institut Curie | Cat#: A-P-R#06 | (dilution 1:100) |
| Antibody | anti-α-Tubulin (mouse monoclonal) | Sigma Aldrich | Sigma-Aldrich Cat# T5168, RRID:AB_477579 | (dilution 1:1000) |
| Antibody | Anti-mouse AlexaFluor 546 F(ab')2 fragment of IgG (H + L) (goat polyclonal) | Life Technologies | | (dilution 1:1000) |
| Sequence-based reagent | gRNA-3-mRAB6A | Eurofins | sgRNA | GTCTCCGC CCGTGGAC ATTG |
| Chemical compound, drug | Nocodazole | Sigma Aldrich | M1404 | (0.1 μM) |
| Chemical compound, drug | Golgicide A | Sigma Aldrich | G0923 | (35 μM) |
| Chemical compound, drug | Taxol | Sigma Aldrich | T7402 | (0.1 μM) |
| Chemical compound, drug | Biotin | Sigma Aldrich | B4501 | (40 μM) |
| Chemical compound, drug | Poly-L-lysine | Sigma Aldrich | P8920 | (0.01% diluted in water) |
| Chemical compound, drug | Fibronectin | Sigma Aldrich | F1141 | (2 μg/mL; 10 μg/mL; 20 μg/mL) |
| Chemical compound, drug | PLL-g-PEG | Surface Solutions | PLL(20)-g[3.5]-PEG(2) | (100 μg/mL) |
| Chemical compound, drug | azido-PLL-g-PEG (APP) | https://doi.org/10.1002/adma.201204474 | | (100 μg/mL) |
| Chemical compound, drug | BCN-RGD | https://doi.org/10.1002/adma.201204474 | | (20 μM) |
| Software, algorithm | Matlab | MathWorks | RRID:SCR_001622 | |

*Continued on next page*

*Continued*

| Reagent type (species) or resource | Designation | Source or reference | Identifiers | Additional information |
|---|---|---|---|---|
| Software, algorithm | Fiji, ImageJ | https://doi.org/10.1038/nmeth.2019 | RRID:SCR_002285 | |
| Software, algorithm | Trackmate | https://doi.org/10.1016/j.ymeth.2016.09.016https://doi.org/10.1016/j.ymeth.2016.09.016 | | |
| Software, algorithm | MetaMorph | Molecular Devices | RRID:SCR_002368 | |
| Other | Hoechst 33,342 | Thermo Fisher Scientific | H3570 | (1 µg/mL) |
| Other | DAPI stain | Merck | D9542 | (1 µg/mL) |

## Cell culture

hTERT RPE1 cells (identity authenticated by STR profiling) (CRL-4000 strain, ATCC, Manassas, VA) were cultured at 37°C with 5% $CO_2$ in Dulbecco's modified Eagle's/F-12 medium (Gibco, Thermo Fisher Scientific, Waltham, MA) supplemented with 10% fetal bovine serum (Gibco, Thermo Fisher Scientific, Waltham, MA), GlutaMAX (2 mM) (Gibco, Thermo Fisher Scientific, Waltham, MA), and penicillin (100 U/mL)-streptomycin (0.1 mg/mL) (Gibco, Thermo Fisher Scientific, Waltham, MA). HeLa cells (CCL-2 strain, ATCC, Manassas, VA) were cultured at 37°C with 5% CO2 in Dulbecco's modified Eagle's medium (Gibco, Thermo Fisher Scientific, Waltham, MA) supplemented with 10% fetal bovine serum (Gibco, Thermo Fisher Scientific, Waltham, MA), GlutaMAX (2 mM) (Gibco, Thermo Fisher Scientific, Waltham, MA), and penicillin (100 U/mL)-streptomycin (0.1 mg/mL) (Gibco, Thermo Fisher Scientific, Waltham, MA). Cells were passaged twice a week in a ratio of 1/10 by washing them with PBS (1×) solution and dissociating using TrypLE Express (Gibco, Thermo Fisher Scientific, Waltham, MA) reagent for 5 min. All cell lines were regularly tested for mycoplasma, and only used if mycoplasma was not detected.

## Plasmids, transfection, and stable cell lines

### Plasmids

pLL7.0: hITSN1(1159–1509)-tgRFPt-SSPB WT (Plasmid no. 60419), pLL7.0: Venus-iLID-CAAX (from KRas4B) (Plasmid no. 60411), pMD2.G (Plasmid no. 12259), psPAX2 (Plasmid no. 12260) lentiviral plasmids were bought from Addgene (Watertown, MA). pHR: myr-iRFP plasmid was a gift from Simon de Beco (Institut Curie, France) and pIRESneo3: Str-KDEL-SBP-EGFP-COL10A1 was a gift from Gaëlle Boncompain (Institut Curie, France).

### Transfection

Transfections were performed using X-tremeGENE 9 (Roche Applied Science, Penzburg, Bavaria, Germany) according to the manufacturer's instructions using an equal amount of plasmid DNA for each construct (1 µg) and a ratio of 3:1 of transfection reagent and DNA.

### Stable cell lines

Stable cell lines were generated using two techniques – lentiviral infection and CRISPR cell line development.

### Lentiviral

For lentivirus production, packaging cell line HEK 293T cells were cotransfected with pMD2g (envelope), psPAX2 (packaging) and lentiviral (transfer*) plasmids in a 1:3:4 ratio, respectively (*pHR-, pLVX- or pLL7-based plasmids were used as transfer plasmids). Lentivirus was harvested 48 hr after transfection and filtered from the supernatant of cell culture by passing it through 0.45 µm filter using a syringe. Next, the target RPE1 cell line was transduced for 24 hr with media containing lentiviral particles. Subsequently, RPE1 cells were selected by fluorescence-activated cell sorting (FACS) according to the fluorescence level of transduced protein.

## CRISPR

A CRISPR approach was used to develop a cell line with a heterozygous iRFP-Rab6A knock-in as a Golgi complex label. CRISPR sgRNAs were designed using the optimized CRISPR design tool CRISPOR TEFOR (for sequences see the table below). For sgRNA-encoding plasmids, single-stranded oligonucleotides (Eurofins Genomics, Germany) containing the guide sequence of the sgRNAs were annealed, phosphorylated, and ligated into BbsI site in px335 plasmid, coding for Cas9 (kindly provided by M. Wassef, Curie Institute, Paris, France). Homology arms of ~800 bp were amplified from genomic DNA using PCR primers with 40-bp overhangs compatible with pUC19 backbone digested with Xba1 and Ecor1 (New England Biolabs, Ipswich, MA) (sequences in the table below). Gibson reactions were performed using a standard protocol with home-made enzyme mix (*Gibson et al., 2009*). RPE1 cells were transfected with 90 µL of polyethylenimine (PEI MAX no. 24,765 Polysciences, Warrington, PA) and 15 µg of the pX335-gRNA and pUC19-homology arms-iRFP plasmids, both diluted in 240 µL NaCl 150 mM. And 7 days after transfection, positive cells were sorted with FACS for enrichment, after additional 10 days, FACS sorted again by single cell per well in a 96-well plate. All 96 clones were screened by PCR and 8 clones were selected for further verification by Western blotting, followed by sequencing.

| Name | Guide sequence (5'–3') |
|---|---|
| gRNA-3-mRAB6A | GTCTCCGCCCGTGGACATTG |
| LeftArm fwd | gaccatgattacgccaagcttgcatgcctgcaggtcgactGCCACAGTGCTCCGCTTTCC |
| LeftArm rev | gcgacggatccttcagccatTGTGGAACTAGAGGAGCGGC |
| Linker-iRFP fwd | gccgctcctctagttccacaATGGCTGAAGGATCCGTCGC |
| Linker-iRFP rev | ccgaagtctgcgcgcgtggaCCGGATTGGCCACTCTTCCAT |
| RightArm fwd | tggaagagtggccaatccggTCCACGgGaGgAGACTTCGG |
| RightArm rev | GggttttcccagtcacgacgttgtaaaacgacggccagtgCAGTGATGAAAGTCAAGAGAACAAAATG AGGTTTTCCG |

## Micropatterning

### Coverslip preparation

Coverslips for live cell imaging were prepared by cleaning round glass coverslips (d = 25 mm, 0.17 mm thickness) (Menzel Gläser, Thermo Fisher Scientific, Waltham, MA) for 1 min in $O_2$ plasma and incubating them with fibronectin (2 µg/mL) (Sigma-Aldrich, St.Louis, MO) in 100 mM $NaHCO_3$ (pH 8.5) for 1 hr in room temperature. Coverslips were washed with PBS (1×) three times and stored in +4°C in PBS (1×).

### Static pattern

Micropatterned coverslips were prepared as described by *Azioune et al., 2009*: $O_2$ plasma-cleaned coverslips were incubated with 100 µg/mL of PLL-g-PEG (Surface Solutions, Switzerland) in 10 mM HEPES, pH 7.4 for 1 hr. They were then exposed to deep UV through micropatterned quartz/chrome photomasks (Toppan, Round Rock, TX) for 5 min, and incubated with fibronectin (20 µg/mL) in 100 mM $NaHCO_3$ (pH 8.5) for 1 hr.

### Releasable (dynamic) patterns

Releasable micropatterns were prepared similarly, with PLL-g-PEG being replaced by azido-PLL-g-PEG (APP) at 100 µg/mL and fibronectin used at lower concentration (10 µg/mL). Migration was released by addition of 20 µM BCN-RGD for 10 min (described in *van Dongen et al., 2013*).

## Drug assays

In drug assays for 60× resolution experiments, RPE1 cells were treated with NZ (0.1 µM) (Sigma-Aldrich, St.Louis, MO) in complete DMEM/F-12 medium at 37°C in 5% $CO_2$ for the duration of the experiment. Also 30 min before the experiment and before addition of the drug, cells were incubated

with Hoechst 33,342 (1 µg/ml) dye (Thermo Fisher Scientific, Waltham, MA), to label cell nuclei. Then, the dye was washed with 1× PBS buffer (pH = 7.5) and a drug, diluted in complete medium, was added. Cells were imaged immediately after addition of a drug.

In drug assays for Cytonote 6 W experiments, RPE1 cells were treated with NZ (0.1 µM) (Sigma-Aldrich, St.Louis, MO), Golgicide A (0.7 µM, 7 µM, 10 µM, 14 µM, 35 µM and 70 µM) (Sigma-Aldrich, St.Louis, MO), or Taxol (Paclitaxel) (0.1 µM) (Sigma-Aldrich, St.Louis, MO) in complete DMEM/F-12 medium at 37°C in 5% $CO_2$ for the duration of the experiment.

## RUSH and SPI assays

### RUSH assay

RUSH assay was performed as described in *Boncompain et al., 2012* and (*Boncompain and Perez, 2012*). RPE1 cells were transfected with 2 µg of plasmid containing the RUSH system (Str-KDEL-SBP-EGFP-Col10A1) and a GFP-labeled collagen type X (ColX) cargo. And 24 hr after transfection, cells were put on anti-GFP antibody coated glass coverslips and let to attach. After 2 hr, biotin was added (40 µM final concentration from 4 mM stock) to the full medium, triggering the release of the cargo. Cells were imaged for 2 hr, until the ColX cargo passed from the endoplasmic reticulum (ER) to the Golgi complex and then was secreted. Coverslip being covered with anti-GFP antibodies enabled the GFP-labeled ColX cargo capture upon secretion.

### SPI assay

SPI assay was performed as described in *Fourriere et al., 2019*. Round glass coverslips (d = 25 mm) were either autoclaved and incubated in bicarbonate buffer (0.1 M pH 9.5) for 1 hr at 37°C (300 µL, upside down) or plasma cleaned (2 min vacuum, 1 min plasma). Next, the coverslips were transferred to poly-L-lysine (0.01% diluted in water) and incubated for 1 hr at 37°C (300 µL, upside down). After being washed in 1× PBS and dried, they were transferred to a solution of anti-GFP antibodies (diluted in bicarbonate buffer) and incubated for 3 hr at 37°C (70 µL, upside down). After another wash with PBS, cells were seeded on top of coated coverslips in complete medium (at least 2 hr given for cells to attach before conducting a RUSH assay). Antibodies used for coating in this study were rabbit anti-GFP (A-P-R 06; Recombinant Antibody Platform of the Institut Curie; dilution 1:100).

## Optogenetics

For local subcellular activation of a RhoGTPase Cdc42, an optogenetic dimer iLID-SspB was used as described in *Guntas et al., 2015*. It was activated by illumination with blue light (440 ± 10 nm). RPE1 cells used in the optogenetic experiments were engineered to stably express the optogenetic dimer and selected for average-high fluorescence level by FACS (the highest expressing cells were discarded, as they were not responsive to optogenetic activation). Experiments were performed in live cell imaging conditions described in the paragraph 'Imaging', using a digital micromirror device (DMD) projector and a blue (440 ± 10 nm) LED illumination source. The projection of blue light was controlled with an interface of a Matlab script and a microscope controlling MetaMorph software by sending a static pattern of light, or using the imaging routine described below. The illumination pattern was optimized for a local signal and weak illumination to reduce the phototoxicity and enable long-term experiments.

## Immunofluorescence

Cells grown on fibronectin covered coverslips were initially fixed with cold (−20° C) methanol (Carlo Erba Reagents, Val-de-Reuil, France) and afterward rehydrated with PBST (PBS-Tween) buffer (×3). Coverslips were blocked with 3% BSA (Sigma-Aldrich, St. Louis, MO) in PBST buffer for 1 hr. Antibodies against α-Tubulin (mouse, T5168 Clone B-5-1-2, dilution 1:1000; 40 min incubation; Sigma-Aldrich, St.Louis, MO) and secondary antibodies against mouse (AlexaFluor 546 F(ab')2 fragment of goat anti mouse IgG (H + L), dilution 1:1000; 30 min incubation; Life Technologies, Carlsbad, CA) were used for immunofluorescence (IF) staining. Coverslips were incubated with DAPI (1 µg/mL, Merck, Kenilworth, NJ) for 15 min to stain cell nuclei. All washes were done with a PBST buffer. Afterward, coverslips were drained on a Kimwipe (Kimtech, Irving, TX) and mounted on Fluoromount (Sigma-Aldrich, St.Louis, MO) before being imaged.

## Imaging

### Live cell imaging

All imaging except MT dynamics experiments was performed at 37° C in 5% $CO_2$ with an IX71 inverted fluorescence and differential interference contrast (DIC) microscope (Olympus, Melville, NY), controlled with MetaMorph software (Molecular Devices, Eugene, OR). The microscope was equipped with a 60× objective (NA = 1.45), motorized stage and filter wheel with SmartShutter Lambda 10–3 control system (Sutter Instrument Company, Novato, CA), a stage-top incubation chamber with temperature and $CO_2$ control (Pecon, Meyer Instruments, Houston, TX), ORCA-Flash4.0 V3 Digital CMOS camera (Hamamatsu Photonics K.K., Japan), z-axis guiding piezo motor (PI, Karlsruhe, Germany), CRISP autofocus system (ASI, Eugene, OR), a laser control system with azimuthal total internal reflection microscopy (TIRF) configuration (iLas2, Roper Scientific, Tucson, AZ), and a DMD pattern projection device (DLP Light Crafter, Texas instruments, Dalas, TX), illuminated with a SPECTRA light engine (Lumencor, Beaverton, OR) at 440 ± 10 nm. Before imaging, cells were dissociated using Versene Solution (Thermo Fisher Scientific, Waltham, MA) and seeded for adhesion on previously mentioned prepared coverslips.

Live cell imaging for MT dynamics experiments was performed at 37° C in 5% $CO_2$ with an IX71 inverted fluorescence microscope (Olympus, Melville, NY) with a spinning disk confocal unit CSU-X1 (Yokogawa Electric Corporation, Tokyo, Japan), controlled with MetaMorph software (Molecular Devices, Eugene, OR). The microscope was equipped with a 100× objective (NA = 1.45), motorized stage, and filter wheel with SmartShutter Lambda 10–3 control system (Sutter Instrument Company, Novato, CA), a stage-top incubation chamber with temperature and $CO_2$ control (Tokai Hit, Shizuoka, Japan), Prime BSI Scientific CMOS camera (Teledyne Photometrics, Tucson, AZ), z-axis guiding piezo motor (PI, Karlsruhe, Germany), and a laser control system (Gataca Systems, Massy, France). Cells were imaged adhered to a Fluorodish (World Precision Instruments, Sarasota, FL) coated with fibronectin as previously mentioned.

### TIRF microscopy

TIRF was used to excite a thin band of fluorophores close to the membrane of adherent cells and avoid out-of-focus fluorescence (*Mattheyses et al., 2010*). A variation of TIRF, called azimuthal TIRF, was used to generate homogeneous illumination and to avoid fringe interferences and imaging artefacts.

### HILO Microscopy

HILO microscopy was used to illuminate the cell at an angle with a thin inclined beam, which increased the signal-to-noise ratio when imaging the Golgi complex and nuclear markers in the cell.

### Cytonote 6W live cell imaging

Cells were imaged with a lensless Cytonote 6 W imaging system (IPRASENSE, Montpellier, France) in a glass-bottomed 6-well (No. 1.5, Cellvis, Mountain View, CA) placed in an incubator at 37° C in 5% $CO_2$.

### Fixed cell imaging

Static imaging of IF stained MTs was performed with a Leica DMi8 microscope (Leica Camera AG, Wetzlar, Germany) with a CSU-X spinning disk unit (Yokogawa Electric Corporation, Tokyo, Japan), controlled with MetaMorph software (Molecular Devices, Eugene, OR). The microscope was equipped with a 63× oil objective (NA = 1.40), motorized stage, and Rotr filter wheel (Andor, Belfast, UK), ORCA-Flash4.0 V2 Digital CMOS camera (Hamamatsu Photonics K.K., Japan), NanoScanZ z-axis guiding piezo motor (Prior Scientific, Cambridge, UK), and an ILE laser control system with enhanced illumination and imaging upgrade Borealis (Andor, Belfast, UK). Before imaging, cells were fixed following a previously described IF protocol.

## Feedback routine and DMD illumination

### Feedback routine

Imaging feedback routine to follow migrating cells with high magnification (60×) was established by using a combination of scripts in MetaMorph and Matlab. It ensures that the microscope stage moves

together with a moving cell, always keeping it in the field of view. The main script was written in Matlab, which commands MetaMorph through calling its macros called 'journals'. It enables imaging of multiple stage positions (i.e. multiple cells) in multiple wavelengths in one experiment. It can be controlled with a GUI, which displays a selected position and its coordinates and pattern of activation for every cell. The amount of acquisition channels and timing can be selected globally for the full set of cells in the experiment. One specific wavelength is chosen as segmentation channel. The images from this channel are used to segment the shape of the cell and to instruct its position. The segmentation threshold can be adjusted for every position and the watershed algorithms can be chosen to separate two touching objects (i.e. cells) in every case.

## DMD illumination

Local subcellular activation with light for optogenetic experiments was achieved by a DMD (*Davis, 2013*) with dimensions of 640 × 480, able to generate eight-bit grayscale patterns. The pattern was individually adjusted for every cell and dynamically evolved during the experiment according to the cell shape. The activation step was incorporated in the previously described imaging feedback routine.

## Image analysis

### Image segmentation

Live cell imaging data obtained using our cell tracking feedback routine was analyzed using a custom-built Matlab script (which you can find in *Figure 4—source data 1*), which allowed segmentation of the shape of the cell, the Golgi complex, and the nucleus, tracking of their position and visualization of their trajectories. All three structures of interest (cell membrane, the Golgi complex, and the nucleus) were fluorescently labeled, so the shape segmentation was done by using a Matlab function 'graythresh' to select the pixels over a certain threshold of fluorescence intensity. Next, the image was binarized with a function 'im2bw', structures touching the image border were deleted with 'imclearborder', small objects were removed from the image with 'bwareaopen', stuctures were closed by dilate erode with 'imclose', using a disk with a radius of 10 pixels as a structuring element, and the holes in the structure were filled with 'imfill' function. The resulting segmented cell shape was then used to extract the 'centroid' of the cell by 'regionprops' function. The same procedure was used to segment the Golgi complex and the nucleus shapes, but with slightly different threshold range. In addition to that, the centroid search area was optimized by selecting the centroid closest to the centroid in the previous image, which is particularly useful when several regions of interest are found in the image. The extracted centroids of all three structures were then concatenated into trajectories depicting the movement through the full experiment. Using our tracking routine implies that the microscope stage would move when a cell moves out of the defined field of view. In this case, the trajectory presents jumps, due to the stage movement. These jumps were corrected using the recorded positions of the stage and a defined scaling parameter. The corrected real trajectories were then used for further analysis.

### Nucleus tracking (semimanual)

Excitation with blue light had to be avoided in optogenetic experiments because of optogenetic system's sensitivity to blue light, so the nuclei in these experiments were tracked semimanually in the DIC channel, using another custom-made Matlab routine (which you can find in *Figure 2—source data 1*). The estimated center of the nucleus was manually chosen by single clicking on the image, the centroids of the nuclei were recorded, concatenated into a trajectory, and corrected according to the stage movement.

### Cell tracking in cytonote 6w data

Initial videos were preprocessed using Fiji commands Gaussian blur (2.0) and Variance filter (10.0), then a Fiji plug-in Trackmate (*Tinevez et al., 2017*) was used to track cells and extract their trajectories. Parameters used were – DoG detector (diameter – 35 pixels, threshold – 5, subpixel localization), Simple LAP tracker (linking max distance – 15, gap-closing max distance – 15, gap-closing max frame gap - 5). Tracks were filtered with several parameters ('Track displacement' – above 5, 'Duration of track' – above 50, 'Track start' – below 61). Extracted cell trajectories were further analyzed using a custom-made Matlab routine (which you can find in *Figure 4—source data 1*).

## Reorientation plot analysis

Evaluation of the Nucleus-Golgi axis reorientation requires two axes: one going through the centroid of the Nucleus and the centroid of the Golgi and another one going through the center of the Nucleus and the center of the optogenetic activation area. The previously described image segmentation techniques were used to segment the optogenetic activation area. Next, the angle between the two axes was calculated using centroid coordinates and an inverse tangent function 'atan' in Matlab. Then, the angle in radians was wrapped to [0 2pi], using the 'wrapTo2Pi' function and unwrapped with 'unwrap' function. The angle, then, was converted from radians to degrees and plotted in a graph.

## Cumulative plot analysis

The speed of the axis reorientation process in different experimental conditions was compared by plotting the data from the previously described reorientation analysis. The first timepoint, when the angle reached 30° was chosen to delineate that the Nucleus-Golgi axis has reoriented, which gave the timing of the reorientation for each cell. Next, the 'cumsum' function in Matlab was used to get the cumulative sum of how many cells have reoriented at a certain timepoint, which was then normalized to 1 (depicting 'all cells') by dividing by the total number of cells in the dataset.

## Morphodynamic map analysis

Morphodynamic map analysis was based on the similar analysis in *Yang et al., 2016*. The cell displacement was followed from frame to frame, providing information, where cell plasma membrane was protruding and where it was retracting. In practice, the contour of the cell in each frame of the video was equidistantly divided into 100 points, called markers. From one frame to another, the pairing of markers was chosen by minimizing the total square distances between markers at time t and t + dt by testing all possible circular shifts of the contour at t + dt. The position of the Nucleus-Golgi axis, calculated from the Golgi and nucleus centroid positions, was plotted on top of the morphodynamic map, showing which way it was pointing. For further analysis and visualization, each column of the obtained morphodynamic maps could be (circularly) shifted, so that the middle marker always represents the Nucleus-Golgi axis, cell trajectory or the x-axis of the image by using Matlab's 'circshift' function.

## Autocorrelation plot

Autocorrelation data were plotted following the (*Gorelik and Gautreau, 2014*) paper, but adapted from Excel to Matlab.

## Cross-correlation analysis of traffic flow

Image stacks in tiff format were adjusted using Fiji's 'Bleach correction - Histogram matching' function. Then, a previously described morphodynamic map of cell protrusions was recorded for every cell, and a Golgi mask was created using the segmentation algorithm described in the section of 'Image segmentation'. A line was drawn from every one of 100 points of the cell boundary in the morphomap toward the centroid of the Golgi mask. Using Matlab's function 'improfile', the mean of fluorescence intensity was calculated along every line. Using these calculations another morphomap, depicting the secretion pattern, was drawn and cross-correlation between the two morphomaps was calculated using 'xcov' function in Matlab.

## Statistical analysis

Unless otherwise noted, all experimental results are from at least three independent experiments. For data which didn't follow normal distribution, we used a nonparametric Wilcoxon rank sum test (comparing two groups) or Kruskal–Wallis test (for experiments with more than two groups) followed by a post hoc Dunn's multiple comparison test. All tests were performed using Matlab. Statistical details of each experiment can be found in the figure legend. Unless otherwise indicated, error bars represent standard deviation (s.d.).

## Modeling

### Morphodynamic map of a single protrusive event

As a first step, a Matlab code was written to simulate realistic morphodynamic maps for a single protrusive event. Similarly to our data, we considered the movement of 100 markers, $x_i$ with $i \in \{0 \ldots 100\}$, equally distributed along the contour of a synthetic cell. To simulate one protrusive event, we numerically implemented a transfer function such that it was similar to the experimental one reported in *Yamao et al., 2015*. This transfer function characterizes the edge movement following a point-like activity of Cdc42. To build it, four terms were summed: (i) a negative Gaussian function with a relatively large width to account for the initial lateral inhibition, (ii) a positive Gaussian function with a relatively small width to account for the actual local protrusive event, (iii) a positive Gaussian function with a mean moving laterally as a function of time to account for the traveling wave, and (iv) a negative Gaussian function in time centered on 0 to account for the central long-lasting inhibition. All those terms were combined with exponentially decaying function of time to account for the temporal 'fading'. This transfer function was further convolved with a synthetic Cdc42 signal to simulate a realistic protrusion event. We assumed that Cdc42 activity for a single protrusive event could be described by a 'signaling puff' in space and time, made using a product between a Gaussian function (lateral extension of the puff) and an exponentially decaying function of time (duration of the puff). The resulting morphodynamic map then represents the edge dynamics following a single protrusive event (note that what we call a single protrusive event could be the outcome of several protrusions sustained by another feedback – e.g. between actin and Cdc42 or Rac1 activity). The duration of Cdc42 activity was chosen such that the duration of the protrusion event was similar to the one observed in our data. For that, we considered the experimental morphodynamic maps under NZ treatment to better isolate 'unique' rounds of localized protrusive activity. Experimental protrusions are extended over c.a. 10 frames, thus, 50 min. We used this number such that our simulation time frame matches the experimental one. The 'single' protrusion morphodynamic map (thus, corresponding to a typical round of localized protrusive activity) was then normalized such that the integral (sum over space and time) was null (zero mean) to ensure a constant cell area. This morphodynamic map is shown in *Figure 6B*.

### First side of the feedback: effect of the polarity axis on protrusions

Next, we simulated a full morphodynamic map by introducing the random appearance of protrusion over time. We took a total duration of the simulation of 1000 frames, $t_{tot} = 1000$, and the frequency at which protrusions happen was characterized by a probability of appearance per unit of time $P_{rand} = 0.25$. The *Figure 6—figure supplement 1A* illustrates morphodynamic maps for different values of $P$. If for a given time a protrusion event happened, the position of this protrusion event along the contour was chosen either randomly (uniform distribution over all markers of the contour) or in the direction of the polarity axis (see below), given by the position of a specific marker $c_{axis}$ that corresponds to the intersection of the internal polarity axis with the cell contour (whose dynamics are described in the next paragraph). The choice of a polarized protrusion was characterized by the probability $P_{polarized}$ (probability of a randomly placed protrusion is then $1 - P_{polarized}$), which quantifies the strength of the feedback between the polarity axis and biased initiation of protrusions. When $P_{polarized} = 0$, protrusions are happening randomly along the contour, and when $P_{polarized} = 1$ protrusions are always happening in front of the polarity axis (see *Figure 6—figure supplement 1B* for an example). To avoid a nonrealistic perfectly straight movement, we introduced a low level of noise for polarized protrusions, namely the position of protrusion was drawn from a normally distributed function whose mean is $c_{axis}$ and standard deviation is $\sigma_c = 5$ (1/20 of the cell contour). Note that our main result does not depend on the exact value of $\sigma_c$. Indeed, this parameter plays a role only on the asymptotic value of the autocorrelation of direction when $P_{polarized} \sim 1$, but not on the actual characteristic decay time of the autocorrelation function (which characterizes the persistence time, our main observable).

### Natural dynamics of the polarity axis

Next, we explicitly modeled the evolution of the internal polarity axis, $c_{axis}(t)$. In absence of any feedback of protrusions on the polarity axis, we could have assumed that this polarity axis remains fixed. However, under this assumption, if we plug-in the first side of the feedback (biased protrusions), the cell would move straight, which would not be realistic. Thus, we modeled a 'natural' movement of the

polarity axis that would mimic the random evolution of the polarity axis when no protrusions affect it. The closest experimental data to infer this dynamic is found for cells plated on a round pattern (see *Figure 5—figure supplement 1F*). In that situation, the polarity axis moves as a correlated random walk whose span reaches 360° in about 4 hr. We simulated such dynamics by assuming that the instantaneous speed of the polarity axis was set by the derivative of a smoothed random function whose values over time are given by a random number between 0 and 10 times the number of markers. The overall natural evolution of the polarity axis, $c_{\text{axis}}^{\text{natural}}$ , is depicted in *Figure 6—figure supplement 1C* for a timescale comparable to experiments and in *Figure 6—figure supplement 1D* for the total duration of the simulation. The actual model we chose for the natural evolution of the polarity axis is arbitrary, nonetheless the details do not matter for the outcome of our simulation. The only important effect is that the polarity axis gets reoriented randomly in about 4 hr.

## Second side of the feedback: effect of protrusions on the polarity axis

To model the other side of the feedback, namely the reorientation of the polarity axis by protrusions, we assumed that an effective force was attracting the polarity axis toward protrusions. This force was computed as the sum of membrane speed over all markers, $F_{\text{prot}} = \sum_{x_i=0}^{100} v_{x_i}$ . The underlying assumption is that every protruding portion of the cell pulls on the polarity axis with a force proportional to the protrusion speed and does this independently of the respective positions of protrusions with regard to the polarity axis. This effective force is competing with the force corresponding to the natural evolution of the polarity axis, $F_{\text{basal}} = d/dt\left(c_{\text{axis}}^{\text{natural}}\right)$ . The strength of the feedback was implemented by introducing a linear combination between these two forces characterized by a value $\kappa$: $d/dt\left(c_{\text{axis}}\right) = \kappa F_{\text{prot}} + \left(1 - \kappa\right) F_{\text{basal}}$ . When $\kappa = 0$, the polarity axis follows its natural evolution. When $\kappa = 1$, the polarity axis follows the protrusions. This parameter $\kappa$ can be expressed as the relative contribution of the effective force toward protrusions, $\kappa = \left(F_{\text{tot}} - F_{\text{basal}}\right) / \left(F_{prot} - F_{\text{basal}}\right)$ , where $F_{\text{tot}}$ is the total force acting on the polarity axis. The *Figure 6—figure supplement 1E* shows an example of morphodynamic maps with two values of the strength of the feedback. As above, the rule chosen to implement the feedback is arbitrary (we could have taken a metric such that protrusions close to the polarity axis matter more than the distant ones), but our goal was to implement a minimal model integrating the feedback.

## Full model with the two-sided feedback

The two sides of the feedback were then implemented in our simulation to produce realistic morphodynamic maps (see *Figure 6—figure supplement 1F* for an example with a low frequency of protrusions and *Figure 6—figure supplement 1G* for an example with realistic protrusion frequency). These morphodynamic maps were then translated into cell trajectories, by noticing that the instantaneous movement of the cell centroid is simply the sum of all the velocities of the markers along the cell contour. These trajectories were analyzed in the same way as the experimental ones to obtain the autocorrelation of direction from which the characteristic decay time was fitted (persistence time). By varying the strength of the two sides of the feedback, we were able to produce the phase diagram presented in *Figure 6G* ('Persistence time'). For visualization purposes, morphodynamic maps were also used to produce synthetic movies of migrating cells (see *Figure 6—video 1*). These movies were obtained by inverting the process of morphodynamic map quantification in order to evolve an elastic contour over time.

## Protrusion unicity index

The protrusion unicity index was computed as the inverse of the average number of simultaneous protrusions, $1/N_p$ . Cells presenting a single protrusion over time are well polarized and have a unicity index close to one, whereas cells presenting several protruding fronts are multipolar and have a unicity index close to zero. To compute $N_p$ , a sliding window of 10 frames (50 min) was applied on morphodynamic maps and protrusions were segmented based on a threshold (70% of maximal protrusion speed). The number $N_p$ was then obtained as the number of nonconnected segmented objects, and this number was averaged over the whole duration of the simulation (1000 frames) and over 20 realizations. As seen from the *Figure 6G* ('Protrusive unicity'), the main parameter dictating the unicity index is $P_{\text{polarized}}$ (if this probability is high, there will always be a single protrusive activity in front of the polarity axis).

## Alignment index

To evaluate how well the polarity axis aligns with cell movement, an alignment index was constructed by computing the standard deviation of the angle between the polarity axis and the instantaneous direction of movement. In terms of circular statistics, this standard deviation is called the angular dispersion and is defined as:

$$r = \sqrt{\left(\frac{1}{n}\sum_{i=1}^{n}\sin\theta_i\right)^2 + \left(\frac{1}{n}\sum_{i=1}^{n}\cos\theta_i\right)^2}$$

The angular dispersion varies between 0 (uniform dispersion) and 1 (perfect alignment). As seen from the *Figure 6G* ('Alignment index'), the alignment index depends both on $\kappa$ and $P_{\text{polarized}}$. When $P_{\text{polarized}}$ is close to zero, even if $\kappa = 1$, protrusions happen randomly all the time, and the polarity axis does not have time to follow them, thus leading to a low value of alignment.

# Acknowledgements

We thank the Institut Curie Cytometry platform for cell sorting; Remy Fert and Eric Nicolau from the Mechanical Workshop for their technical assistance; Pascal Silberzan and Caroline Giuglaris for the access and help with Cytonote 6 W microscope; Gaëlle Boncompain for sharing the RUSH system contructs;; Mathieu Deygas for RPE1::EGFP-α-tubulin cell line; Krek lab for RPE1::EB3-EGFP cell line; Aastha Mathur for advice on the Immunofluorescence (IF) staining protocol; Pallavi Mathur for help with Western blot (WB) technique; Bruno Goud for critical reading of the manuscript and helpful discussions; John Manzi and Fahima Di Federico from UMR168 BMBC platform, Maud Bongaerts and Laurence Vaslin for experimental advice and help with plasmid constructs; Jean de Seze for fruitful discussions and help with adaptation of the cell tracking routine.

# Additional information

## Funding

| Funder | Grant reference number | Author |
|---|---|---|
| Sorbonne Université | Programme doctoral Interface pour le Vivant | Kotryna Vaidžiulytė |
| Fondation pour la Recherche Médicale | FDT201904008167 | Kotryna Vaidžiulytė |
| Labex CelTisPhyBio | ANR-10-LBX-0038 | Kristine Schauer Mathieu Coppey |
| Labex and Equipex IPGG | ANR-10-NANO0207 | Mathieu Coppey |
| Idex Paris Science et Lettres | ANR-10-IDEX-0001-02 PSL | Kristine Schauer Mathieu Coppey |
| Centre National de la Recherche Scientifique | | Kristine Schauer Mathieu Coppey |
| Institut Curie | | Kristine Schauer Mathieu Coppey |
| French National Research Infrastructure France-BioImaging | ANR-10-INBS-04 | Anne-Sophie Macé Mathieu Coppey |
| Institut Convergences Q-life | ANR-17-CONV-0005 | Mathieu Coppey |

The funders had no role in study design, data collection and interpretation, or the decision to submit the work for publication.

## Author contributions

Kotryna Vaidžiulytė, Conceptualization, Data curation, Formal analysis, Funding acquisition, Investigation, Methodology, Project administration, Software, Validation, Visualization, Writing – original draft, Writing – review and editing; Anne-Sophie Macé, Software, Visualization, Writing – review and

editing; Aude Battistella, Resources; William Beng, Software; Kristine Schauer, Conceptualization, Funding acquisition, Methodology, Project administration, Supervision, Writing – review and editing; Mathieu Coppey, Conceptualization, Data curation, Formal analysis, Funding acquisition, Methodology, Modelling, Project administration, Software, Supervision, Validation, Visualization, Writing – original draft, Writing – review and editing

## Author ORCIDs
Kotryna Vaidžiulytė ⓘD http://orcid.org/0000-0002-2114-3612
Kristine Schauer ⓘD http://orcid.org/0000-0002-6102-8790
Mathieu Coppey ⓘD http://orcid.org/0000-0001-8924-3233

## Decision letter and Author response
Decision letter https://doi.org/10.7554/eLife.69229.sa1
Author response https://doi.org/10.7554/eLife.69229.sa2

---

# Additional files

## Supplementary files
• Transparent reporting form

## Data availability
Source data files with numerical data and Source Code for all the graphs in the figures are provided as a zip supplementary file attached to each figure in this submission. Raw imaging data for all the figures are available in the BioImage Archive repository at https://www.ebi.ac.uk/biostudies/studies/S-BIAD365 with BioStudies accession number S-BIAD365.

The following dataset was generated:

| Author(s) | Year | Dataset title | Dataset URL | Database and Identifier |
|---|---|---|---|---|
| Coppey M | 2022 | Persistent cell migration emerges from a coupling between protrusion dynamics and polarized trafficking | https://www.ebi.ac.uk/biostudies/studies/S-BIAD365 | BioImage Archive, S-BIAD365 |

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
