## [Editor Report]

It has previously been suspected that secretion supports cell migration in human cells. This study proposes a physical model and offers various results that elegantly link the activation of a small GTPAse at the leading edge with the re-organisation of the secretory pathway, creating a feedback loop that allows persistence of direction. Hopefully, the simple physical model will serve as a foundation to include more regulatory loops in the conceptualisation of cell migration.

---

## [Decision Letter]

**Decision letter after peer review:**

Thank you for submitting your article "Persistent cell migration emerges from a coupling between protrusion dynamics and polarized trafficking" for consideration by *eLife*. Your article has been reviewed by 3 peer reviewers, including Frederic A Bard as Reviewing Editor and Reviewer #1, and the evaluation has been overseen by Jonathan Cooper as the Senior Editor. The following individual involved in review of your submission has agreed to reveal their identity: Leah Edelstein-Keshet (Reviewer #2).

In this paper, Vaidziulyte et al., study the interaction between the alignment of the Nucleus-Golgi (NG) axis with the GTPase-directed protrusive activity of cells. They explore the bidirectional link between directed secretion from the Golgi (the Nucleus-Golgi polarity axis), and events on the cell membrane associated with protrusive activity. They label the Golgi complex and track migrating cells, showing that the Nucleus-Golgi axis aligns to the direction of motion.

The authors treat cells with microtubule-disrupting nocodazol (NZ), finding decreased migratory persistence. Using maps of morphodynamic and of Rab6-labeled Golgi secretion (trafficking maps), they find that protrusion precedes trafficking. They optogentically stimulate protrusion by activating Cdc42, showing downstream reorientation of the nuclear-golgi axis that is faster in circular confined cells that in free-moving cells. Finally, the authors describe a minimal model to fit their data to two parameters that govern the feedback between the axis of polarity and the protrusive activity.

The strengths of the paper are quality imaging, quantitative analysis, the use of optogenetics and the interesting tracking of cells with and without confinement, with and without MT disruption.

The limitations of the paper are the lack of integration of the microtubule network in their analysis and model, while the re-organisation of this network is an obvious link between protrusions and Golgi-Nucleus axis re-orientation.

The authors have attempted to isolate the Nucleus-Golgi axis as an important factor, with no direct demonstration of the role of Golgi derived secretion. They also have not entirely evaluated its importance relative to other factors. For example, could the NZ treated cells simply have distinct GEF/GAP activities? Is the lack of persistence in such cells explainable only by trafficking defect?

Finally, the model is very minimal, which can be an advantage (only 2 parameters needed to fit the data). At the same time this minimality also means that there is no clear mechanistic hypothesis to test, other than the relatively well known fact that protrusion and polarity feed back on one another. Nowhere in the model is the secretory property of the Golgi, or indeed any specific property of the NG axis used. In short, the "axis" could just as easily relate to any other structural cell property that responds to force.

While the paper is interesting and commendable, it has 3 important limitations that will need to be addressed:

1) The microtubule network is only indirectly incorporated, through the use of Nocodazole. For instance, microtubules are not imaged. This leads to a major blind spot in the description of the phenomenon of persistent cell migration. This is apparent in Figure 4A, where re-organisation of secretion appears to precede reorientation of the Nucleus-Golgi axis, but is not much discussed. It is also apparent in the Discussion section. The sentence "We demonstrated using optogenetics that sustained local activation of Cdc42 is sufficient to reorient the Nucleus-Golgi axis in 2-4 hours" somewhat suggests that reorientation of the axis could occur independently of the microtubule network, which is highly unlikely. It is critical that the authors incorporate more data and more discussion about the role of the microtubule network.

2) The authors do not demonstrate directly that Golgi-derived secretion mediates the persistence of migration; it is only inferred through reference to previous work. A more direct confirmation, for instance with Rab6 knockdown or dominant-negative, would enhance the thesis of the manuscript. While we recognize that this caveat is discussed by the authors, the fact that the paper is highly focused on the Golgi-Nucleus axis is at odds with the lack of a direct test of the importance of Golgi-derived secretion.

3) As noted by two reviewers, the model appears rather minimalist and its predictive value is not clear. A clear set of predictions would greatly enhance the value of the model. At a minimum, a clear path for the development of a more predictive model should be discussed.

Please review the other comments below to address them.

*Reviewer #1 (Recommendations for the authors):*

I have the following recommendations:

1) The authors should highlight more clearly how their study advance the conceptual understanding of persistent cell migration.

2) The authors should discuss in more detail the role of the microtubule network in linking protrusions and polarised trafficking. In particular, discuss how the literature on Cdc42 and microtubules can complement their model. Figure 4A is of particular interest to this reviewer. Indeed, the authors show that the re-organisation of secretion precedes the reorientation of the Golgi, consistent with the fact the microtubules directing secretion re-organise before the Golgi is re-oriented. This should be explored further. At a minimum, the authors should show the images of the Golgi and nucleus (not just the RUSH cargo) in figure 4A. The time-line of protrusion redirection, secretion re-orientation and Golgi re-orientation could be precisely measured using the optogenetic tool, although I realise it might be challenging to get all the constructs required in the same cells.

3) The authors should discuss more how their model differs from previous models and whether it allows more predictions. The authors could also discuss how the parameters they defined: Ppolarised and K may relate to or integrate molecular events.

Other points:

– Figures pages are not labelled, making it a bit confusing for reviewers.

– How is tEscape in Figure 2 defined is not described.

*Reviewer #2 (Recommendations for the authors):*

My recommendations are as follows:

(1) To help rule out the NZ treatment as merely a change in the GTPase- associated GEF/GAP profiles, I.e. to link it more directly to secretion from Golgi, and (2) To consider a more mechanistic description of the process, and show that, under appropriate limits or assumptions, it can be reduced to your minimal model. This may help to link the model parameters to experimental manipulations in the future.

*Reviewer #3 (Recommendations for the authors):*

1) the different values of the angles quantified should be written with SD (e.q., value of angle difference between direction and Golgi-Nucleus 0+/- 30{degree sign}).

2) For each diagram, please precise the axis used: direction of migration or Nucleus-Golgi axis…

3) In Figure 3G, cell tracks are difficult to observe in presence of nocodazole. Does it mean that cells are turning on themselves? How the authors could determine a speed of migration if there is no displacement of the cell body?

4) Could the authors comment why there is two phases of decay in the correlations curves presented in Figure 3H? Indeed, the data presented in Figure 6F (3) proposed that this correlation could be stable over time and not only decreasing.

5) The authors should provide statistic about the number of cells accumulating secreted cargos in the newly formed protrusions.

6) MT dynamics is essential to sustain secretion mechanisms that were observed with secretion of Collagen X through the RUSH system or following Rab6 vesicles outside the Golgi apparatus. However, this section of the manuscript is not extremely clear despite being highly interesting. Precise correlation analysis and quantification of characteristic time for each processes (Collagen X secretion and Rab6 reorganization) will be highly beneficial to the study after comparison with the dynamics of nucleus-Golgi reorganization. Moreover, this is not clear how local CDC42 activation can affect these two parameters to sustain migration persistency.

7) What is the consequence of the use of a Rab6 dominant negative on the correlation between cell protrusions, migration persistency and nucleus-Golgi reorientation?

8) It would have been interesting to test if the activation of another GTPase inducing cell protrusion as Rac1 could also induce nucleus-Golgi reorganization qualitatively or in the same temporal range. It is very impressive and surprising that optogenetic activation of Cdc42 could also induced nucleus-Golgi reorientation when MT dynamics is perturbed. Indeed, the author did not comment precisely in the discussion how this mechanism could occurs.

9) The authors should also indicates non-persistency and super-persistency behaviors (norma-persistency will be also interesting) and that the black lines are experimental data, both in Figure 6G.

10) The physical model is very elegant despite being poorly and simply explained in a clear scheme that should be proposed in the Figure 6. Indeed, all the elements are accessible but widely spread in the main text and supplemental material. A scheme of this models and the different interplays involved will help its understanding to a wide readership.

---

## [Author Response]

The reviewers have discussed their reviews with one another, and the Reviewing Editor has drafted this to help you prepare a revised submission.In this paper, Vaidziulyte et al., study the interaction between the alignment of the Nucleus-Golgi (NG) axis with the GTPase-directed protrusive activity of cells. They explore the bidirectional link between directed secretion from the Golgi (the Nucleus-Golgi polarity axis), and events on the cell membrane associated with protrusive activity. They label the Golgi complex and track migrating cells, showing that the Nucleus-Golgi axis aligns to the direction of motion.The authors treat cells with microtubule-disrupting nocodazol (NZ), finding decreased migratory persistence. Using maps of morphodynamic and of Rab6-labeled Golgi secretion (trafficking maps), they find that protrusion precedes trafficking. They optogentically stimulate protrusion by activating Cdc42, showing downstream reorientation of the nuclear-golgi axis that is faster in circular confined cells that in free-moving cells. Finally, the authors describe a minimal model to fit their data to two parameters that govern the feedback between the axis of polarity and the protrusive activity.The strengths of the paper are quality imaging, quantitative analysis, the use of optogenetics and the interesting tracking of cells with and without confinement, with and without MT disruption.

Thank you for the appreciation of our work.

The limitations of the paper are the lack of integration of the microtubule network in their analysis and model, while the re-organisation of this network is an obvious link between protrusions and Golgi-Nucleus axis re-orientation.

We have now better integrated the MT network in our manuscript, providing new data on MT modifications during drug treatment and discussing the role of MTs in the feedback between protrusions and internal polarity axis.

The authors have attempted to isolate the Nucleus-Golgi axis as an important factor, with no direct demonstration of the role of Golgi derived secretion. They also have not entirely evaluated its importance relative to other factors. For example, could the NZ treated cells simply have distinct GEF/GAP activities? Is the lack of persistence in such cells explainable only by trafficking defect?

As specified in the text, we isolated the Nucleus-Golgi axis as a reliable proxy for persistent migration, but not as a determinant factor. Regarding the direct demonstration of Golgi-derived secretion, we have added new data (Figure 4—figure supplement 3 and 4) on Golgi-based trafficking and have evaluated its importance relative to the microtubule network organization. We conclude from our previous data and new results that Golgi-based polarized secretion partially accounts for persistent migration, thus other factors are required, as suggested by reviewers. While it would be very interesting to identify these other factors and establish their relative contributions, we think that this goes beyond the scope of our work.

Regarding the GEF/GAP activities, we cannot rule out that NZ does not change GEF/GAP activities that regulate the activation level of Cdc42. As extensive GEFs/GAPs have been described for Cdc42, it will be very challenging to investigate this in detail. However, we think that this effect is limited in our experimental conditions (low dose, 0.1 µM of NZ) given the fact that the speed of protrusions stays the same in NZ treated cells as in WT cells (Figure 3C). If there was a strong effect of NZ on RhoGTPases’ activities, it should be visible on protrusion dynamics. We added a sentence in the discussion on p. 16, lines 358-360 to highlight this point.

Finally, the model is very minimal, which can be an advantage (only 2 parameters needed to fit the data). At the same time this minimality also means that there is no clear mechanistic hypothesis to test, other than the relatively well known fact that protrusion and polarity feed back on one another. Nowhere in the model is the secretory property of the Golgi, or indeed any specific property of the NG axis used. In short, the "axis" could just as easily relate to any other structural cell property that responds to force.

We indeed intended to build a very minimalistic model. We do recognize that the issue is then to demonstrate how useful such a model is. We think that our model brings something more substantial than the known fact of protrusion and polarity feeding-back on one another, namely its quantitative aspect: the model can be used retrospectively to quantify the contribution of mechanisms targeted by different experimental conditions or found in different cells for persistent migration (see new data on NZ, Golgicide A and HeLa cells Figure 6—figure supplement 2 and 3). It also can be used predictively to quantitatively change the persistence time of migration by modulating the feedback strengths. As with any quantitative model, if the model fails to quantitatively fit either retrospections or predictions, it will inform us on the fact that a piece of information is missing, and in that case the model serves as a tool to progress our understanding.

Regarding the notion of “axis” in our model, we added a clarification on this notion in the discussion (p.16-17). In our model we do not refer to a particular structural cell property, our intention was to develop a minimal coarse-grained picture that does not rely on a particular structure. However, when we made use of our experimental data to assess the quantitative aspect of our model, we explicitly used the data of the NG axis (for the calculation of the alignment). Given that our data quantitatively fit together in the context of our minimal model, it suggests that indeed the dynamics of the secretion axis, and NG axis as a proxy, are relevant for persistent migration. Yet, since many other structural properties may correlate with the NG axis, this is not a proof that this axis is the only existing proxy for cell polarity.

While the paper is interesting and commendable, it has 3 important limitations that will need to be addressed:1) The microtubule network is only indirectly incorporated, through the use of Nocodazole. For instance, microtubules are not imaged. This leads to a major blind spot in the description of the phenomenon of persistent cell migration. This is apparent in Figure 4A, where re-organisation of secretion appears to precede reorientation of the Nucleus-Golgi axis, but is not much discussed. It is also apparent in the Discussion section. The sentence "We demonstrated using optogenetics that sustained local activation of Cdc42 is sufficient to reorient the Nucleus-Golgi axis in 2-4 hours" somewhat suggests that reorientation of the axis could occur independently of the microtubule network, which is highly unlikely. It is critical that the authors incorporate more data and more discussion about the role of the microtubule network.

We agree with the raised comments, and we have now added additional data revealing the MT network in different conditions. We have explicitly added a discussion on the delay in Rab6-dependent secretion relative to the reorientation of the Nucleus-Golgi axis as well as clarified that Nucleus-Golgi axis reorganization is MT-dependent (see discussion p. 14-15).

Regarding new data on MT, we have carried out a series of live-imaging experiments in RPE1 cells labelled with different microtubule markers in control conditions and in cells exposed to low Nocodazole concentrations (0.1 µM). We have used stable RPE1 cell lines expressing fluorescent (EGFP) α-Tubulin for labelling the microtubule network, and fluorescent (EGFP) EB3 for labelling the growing microtubule “+” ends. We have also performed immunofluorescence (IF) staining labelling α-Tubulin to investigate the changes of the microtubule network over longer periods of time (1h, 3h, 6h and 24h). The results are summarized in the new Figure 3 —figure supplement 1.

To summarize the obtained results:

1. The IF experiments have demonstrated that with low Nocodazole concentration (0.1 µM) added, the microtubule network is not destroyed even in long periods of time (1h, 3, 6h, and 24h) (see Figure 3 —figure supplement 1A).

2. Following the IF experiment, we have chosen the time point of 3h after exposure to Nocodazole to investigate live dynamics of the microtubule network (see new Figure 3-video 2). The experiments with RPE1-EGFP-Tubulin cells have shown an intact microtubule network, even with exposure to Nocodazole (see Figure 3 —figure supplement 1B).

3. To investigate the microtubule growth dynamics, we have conducted short live-imaging experiments in RPE1-EGFP-EB3 labelled cells at the time point of 3h exposure to Nocodazole (see new Figure 3-video 2). Here, Nocodazole exposed cells have shown slowed down dynamics of microtubule growth, slower turnover, lower persistence and smaller EB3 “comet” size, proposing reduced recruitment of EB3 protein to the growing microtubule “+” ends (see Figure 3 —figure supplement 1C).

These results confirm previous research on the effect of low doses of Nocodazole on microtubule network (see Yvon, Anne-Marie C, and Patricia Wadsworth. “Nanomolar Concentrations of Nocodazole Alter Microtubule Dynamic Instability in vivo and in vitro”, Molecular Biology of the Cell, 1997). Together with our data on secretion (see Figure 4 —figure supplement 1B, showing that secretion is more uniform in cells exposed to Nocodazole), our results show that active microtubule dynamics are required for polarized secretion.

2) The authors do not demonstrate directly that Golgi-derived secretion mediates the persistence of migration; it is only inferred through reference to previous work. A more direct confirmation, for instance with Rab6 knockdown or dominant-negative, would enhance the thesis of the manuscript. While we recognize that this caveat is discussed by the authors, the fact that the paper is highly focused on the Golgi-Nucleus axis is at odds with the lack of a direct test of the importance of Golgi-derived secretion.

Following the advice of referees, we have specifically investigated the role of Golgi-derived secretion and concluded that it plays a partial role in the establishment of persistent migration. We have added corresponding data and further discussed our results (see p. 9 and new Figure 4—figure supplement 3 and 4).

For this, we have used Golgicide that perturbs secretion from the Golgi by specifically targeting GBF1, a GEF of Arf1 for the COPI coat production at the Golgi. We found that treatment with Golgicide A reduces persistence, however less than treatment with either NZ or Taxol (Figure 4—figure supplement 4). Moreover, to specifically investigate the role of Rab6-dependent secretion we have compared a GFP-Rab6 overexpressing cell line with the endogenous iRFP-Rab6 cell line that does not show overexpression of Rab6 (Figure 4—figure supplement 3). Interestingly, we found, that overexpression of Rab6 increases persistence. Altogether, this suggests that secretion from the Golgi participates in the feedback but its contribution is less than that of the microtubule network dynamics and factors/processes associated to it.

We also performed a silencing of Rab6 using a stable cell line expressing a shRNA against Rab6. We did not observe any difference in persistence (see Author response image 1), which could be explained by the presence of compensatory mechanisms that have been acquired during the selection of this stable cell line during which other proteins could take over Rab6 function in polarized secretion.

**Author response image 1. sa2fig1:** 

3) As noted by two reviewers, the model appears rather minimalist and its predictive value is not clear. A clear set of predictions would greatly enhance the value of the model. At a minimum, a clear path for the development of a more predictive model should be discussed.

We agree with the two reviewers that the usefulness of the minimal model was disputable in the previous version of the manuscript. To better exploit our model, we have now improved its use and discussed its predictive power. More specifically:

1. To insist on the quantitative aspect of the model, we have run the model with finer grids to produce better phase diagrams (revised Figure 6 last panel) and we now show the values of persistence time, protrusive unicity, and alignment index with their means and SEM values. We would like to point out here that the fact that our model is quantitatively consistent with the data is not trivial at all. Indeed, the timescale over which cells are persistent is equivalent to the timescale over which cells lose their direction of motion. This loss of directionality can happen either because polarized migrating cells drift slowly in their direction of migration, or because polarity breaks down and a new axis emerges (redirection of motion). Looking at the morphodynamic maps, both effects are happening, and our model seems to capture them with our minimal ingredients.

2. To generalize our model beyond RPE1 cells, we have applied our model to HeLa cells (Figure 6—figure supplement 2). We show that this less persistent cell line also fits quantitatively with our model. We added the result in the Results section, p. 13.

3. To characterize the effect of the NZ and Golgicide on the feedback strengths, we analyzed both datasets (Figure 6—figure supplement 3). We show that both drugs affect the two sides of the feedback, NZ treatment having a greater effect. We added the result in the Results section, p. 13 and 14.

4. We now discuss the predictive power of the model, and we propose future experiments to test it, see discussion p. 17.

Please review the other comments below to address them.Reviewer #1 (Recommendations for the authors):I have the following recommendations:1) The authors should highlight more clearly how their study advance the conceptual understanding of persistent cell migration.

Our study demonstrates that persistent mesenchymal cell migration can be quantitatively described by a simple feedback. While this feedback is not new, its quantitative aspect is. For example, our model shows how the decay time of persistency is connected to the two strengths of the feedback. It is not just about how persistent migration emerges; it is also about how it is limited by the inherent turns that come from the strengths of the two sides of the feedback being less than 1. Along the same line, this quantitative aspect can be used to understand the less persistent behavior of HeLa cells compared to RPE1 cells. Our model suggests that HeLa cells are less persistent mostly because Ppolarized is smaller. This small value of Ppolarized can be recognized by the multiple competing fronts observed in these cells. Regarding the conceptual understanding, our study supports the idea that persistent migration can be quantitatively captured at the level of coordination of high-level functions (polarity, secretion, protrusions) rather than solely attributed to a given specific molecular mechanism. We hope that the revised manuscript will better emphasize those points and better satisfy Reviewer #1.

2) The authors should discuss in more detail the role of the microtubule network in linking protrusions and polarised trafficking. In particular, discuss how the literature on Cdc42 and microtubules can complement their model. Figure 4A is of particular interest to this reviewer. Indeed, the authors show that the re-organisation of secretion precedes the reorientation of the Golgi, consistent with the fact the microtubules directing secretion re-organise before the Golgi is re-oriented. This should be explored further. At a minimum, the authors should show the images of the Golgi and nucleus (not just the RUSH cargo) in figure 4A.

We have added an additional paragraph to the Discussion section, where we discuss the role of the microtubule network in linking protrusions and polarized trafficking in greater detail (p. 14-15). We have also added the outlines of nucleus and Golgi in Figure 4A, with the original images added in the source files.

The time-line of protrusion redirection, secretion re-orientation and Golgi re-orientation could be precisely measured using the optogenetic tool, although I realise it might be challenging to get all the constructs required in the same cells.

We have tried to perform the experiment several times, but, indeed, it is very hard to have the RUSH and optogenetics constructs on the same cell, as we were limited by the imaging channel incompatibility (in our constructs, the secretory cargo imaging channel is overlapping with one of the optogenetic constructs).

Yet, we can comment on the timescale of different events, while looking at the different experiments we did. The timing of protrusion redirection is not a well-defined quantity for us since we assume that the redirection of cell movement starts with a random protrusion. Thus, for us, protrusion redirection happens as fast as protrusion happen. Yet, if we want to have a timescale, from our optogenetic experiments we see that protrusions are “redirected” within 3-5 minutes. This timing is consistent with the quick relocalization of protrusive activity observed in the morphodynamic maps when cells make quick turns (see for example cells 2, 9, 11, and 14 in Figure 4—figure supplement 2). This timing is implemented in our physical model by the frequency of protrusions per unit of time (new Figure 6D). The timing for secretion redirection was calculated to be ~20min from freely moving cells (Figure 4E). And the timing for Golgi reorientation, ~3 hrs, can be obtained from optogenetic experiments. Altogether, these different time intervals can be ordered, and we have introduced them visually in the new scheme of Figure 6A.

3) The authors should discuss more how their model differs from previous models and whether it allows more predictions.

We have added a comment on the predictive power of the model in the discussion p. 17. Particularly, our model can be used to predict the persistence time of cells from the measurement of two parameters – the average number of competing protrusions and the alignment of polarity axis with direction of motion. Moreover, it allows to estimate how changes in these parameters could make cells e.g. to “super-persisters”.

The authors could also discuss how the parameters they defined: Ppolarised and K may relate to or integrate molecular events.

In our work, we wanted to keep the two parameters as coarse-grained parameters without having to rely on very specific molecular events or proteins of interest. Our aim was to focus on the coordination of high-level functions (polarity, secretion, protrusion) which themselves depend on a huge number of molecular events/actors. Being explicit on these would introduce many unknown parameters and somehow arbitrary rules (there are already a couple of such rules in our minimalist model). Yet, if needed, we think that there are many simple ways by which our parameters could integrate underlying molecular processes. Here is a constructive example of one way to start defining molecularly our parameters. For Ppolarized, let us consider *N* routes of possible cargo secretion. Given the density ρ(χ) of these routes along the contour of the cell, χ being the coordinate along the contour, the probability of secretion at a given point χ will be given by ρ(χ)/N. Then Ppolarized will be completely defined by the density of secretion routes, which then need to be further modelled with other molecular events (focal adhesions, dynamics of MTs, etc.). For κ, let us focus on MTs only. Let us consider that there are M MTs connecting the Golgi and the cell edge, whose local density is m(χ). Now consider a given local density of dynein, specified by the concentration d(χ). The individual force of MT exerted on the Golgi will be proportional to m(χ) × d(χ). For an individual protrusion, the force corresponding to it will be Fprot=∫x∈protrusiondxm(x)×d(x). The force of all the other MTs will be Fbasal=∫x∈notprotrudingdxm(x)×d(x). Then the parameter κ is completely defined, but m(χ) and d(χ) need to be further modelled with other molecular events.

Other points:– Figures pages are not labelled, making it a bit confusing for reviewers.

We are sorry about this inconvenience; we uploaded the figures separately and the *eLife* online system combined them together without page numbering**.**

– How is tEscape in Figure 2 defined is not described.

tEscape is defined in the legend of Figure 2A:

“escape” is defined to be the moment when the center of the cell nucleus is leaving the area of the pattern”.

Reviewer #2 (Recommendations for the authors):My recommendations are as follows:(1) To help rule out the NZ treatment as merely a change in the GTPase- associated GEF/GAP profiles, I.e. to link it more directly to secretion from Golgi, and (2) To consider a more mechanistic description of the process, and show that, under appropriate limits or assumptions, it can be reduced to your minimal model. This may help to link the model parameters to experimental manipulations in the future.

For point (1), see our responses above: p. 2 two first paragraphs, and response to point (2) p. 4. For point (2), see our response above to point (3) of Reviewer #1 p. 6.

Reviewer #3 (Recommendations for the authors):1) The different values of the angles quantified should be written with SD (e.q., value of angle difference between direction and Golgi-Nucleus 0+/- 30{degree sign}).

We have added the values of the angles and their SD for Figure 1C, 5B, 5C and Figure 5—figure supplement 1B in the text (p. 5 and 10).

2) For each diagram, please precise the axis used: direction of migration or Nucleus-Golgi axis…

We have unified our color scheme for diagrams, the Nucleus-Golgi axis is represented by a black arrow and the direction of migration by an orange arrow. We also checked if it was correctly mentioned in all legends.

3) In Figure 3G, cell tracks are difficult to observe in presence of nocodazole. Does it mean that cells are turning on themselves? How the authors could determine a speed of migration if there is no displacement of the cell body?

Author response image 2 is a zoom of the trajectories of Figure 3G, where the scale is approximatively the typical size of a cell. As seen in this graph, NZ treated cells mostly stay in place and jiggle around, but the center of mass is actually moving which enable us to determine a speed of migration. For example, even if a cell moves a tenth of its body length thanks to a given protrusive activity, there is still a displacement of the center of mass and a velocity given accordingly.

4) Could the authors comment why there is two phases of decay in the correlations curves presented in Figure 3H? Indeed, the data presented in Figure 6F (3) proposed that this correlation could be stable over time and not only decreasing.

We are not sure about which phases of decay in the correlation curves the reviewer is talking about. The first instantaneous decay from 1 to 0.8 (in Figure 3H) is due to noise in the positioning of the cell center of mass. Then the decay curve could be interpreted as biphasic, with a possible crossover (change of slope) happening around 2hrs. But this change of slope, if real, is very hard to distinguish from a single exponential decay, and we did not consider it. In the data of Figure 6F (now 6G, third case), there is no stability: what looks like a plateau is not, it is just that the decay time is much longer than the observation window. The curve will eventually go to zero. The other “fast” decay in this curve (from 0 to 20), is due to the “jiggly” pattern of cell movement on short time scale, due to fluctuations of cell shape.

5) The authors should provide statistic about the number of cells accumulating secreted cargos in the newly formed protrusions.

We have added an additional folder in the Source files with all the raw experimental data for the RUSH-SPI secretion experiments, and specified more clearly the numbers in the text of the manuscript (page 8, lines 168-173).

6) MT dynamics is essential to sustain secretion mechanisms that were observed with secretion of Collagen X through the RUSH system or following Rab6 vesicles outside the Golgi apparatus. However, this section of the manuscript is not extremely clear despite being highly interesting. Precise correlation analysis and quantification of characteristic time for each processes (Collagen X secretion and Rab6 reorganization) will be highly beneficial to the study after comparison with the dynamics of nucleus-Golgi reorganization.

For Rab6, we already performed a precise correlation analysis (Figure 4B-E). We agree with reviewer #3 that it would be very good to also have this correlation for the RUSH assay. We have tried our best to quantify the corresponding videos but, unfortunately, without success. The signal of the cargo is changing a lot over time (intensity, spatial distribution) which renders any systematic quantification extremely difficult. The best is to look directly at the videos, from which the reader can appreciate that dynamics of Collagen X secretion, which happens on similar timescales as Rab6 redirection of secretion and Fourriere et al., (2019) have shown that ∼70-80% of vesicles of ColX were positive for RAB6.

Moreover, this is not clear how local CDC42 activation can affect these two parameters to sustain migration persistency.

We show that Cdc42 activation regulates the dynamics of Nucleus-Golgi reorientation (Figure 5). Unfortunately, we were not able to reveal how Cdc42 activation regulates secretion of Collagen X and Rab6-vesicle reorganization due to technical limitations. Indeed, it is very challenging to combine optogenetics that allow us to control Cdc42 activation with the RUSH system that allows to study in detail secretory dynamics due to spectral overlap of the different probes required. Despite of several tries, we were not successful and therefore cannot answer how Cdc42 activation affects secretion.

7) What is the consequence of the use of a Rab6 dominant negative on the correlation between cell protrusions, migration persistency and nucleus-Golgi reorientation?

Because we did not have a cell line that express a dominant negative Rab6, we address this point by comparing the GFP-Rab6 overexpressing cell line with the endogenous iRFP-Rab6 cell line that does not show overexpression of Rab6 used in the study. Interestingly, we found that overexpression of Rab6 increases persistence (see also pour answer to point 2) (p. 4).

8) It would have been interesting to test if the activation of another GTPase inducing cell protrusion as Rac1 could also induce nucleus-Golgi reorganization qualitatively or in the same temporal range.

This experiment was done in H. Hao et al., EMBO Reports 2020 (see their figure 5). The authors show that optogenetic activation of Rac1 induces Nucleus-Golgi reorientation. Yet, this effect could be still the consequence of Cdc42, since both Rac1 and Cdc42 crosstalk to each other (see our previous work, S. De Beco et al., Nature Communication 2018).

It is very impressive and surprising that optogenetic activation of Cdc42 could also induced nucleus-Golgi reorientation when MT dynamics is perturbed. Indeed, the author did not comment precisely in the discussion how this mechanism could occurs.

Indeed, we did not discuss this, because our data are not sufficient to address this point. Since NZ-treatment disturbs the Golgi organization, we cannot correctly estimate the direction of the Nucleus-Golgi axis throughout the experiment. Our data suggest that Cdc42 activation is sufficient for persistent migration, but it does not allow us to reveal whether Golgi is reoriented. Our interpretation of these experiments, supported by our model, is that the Nucleus-Golgi axis in not required in the case where Cdc42 is activated externally (such as by optogenetics).

9) The authors should also indicates non-persistency and super-persistency behaviors (norma-persistency will be also interesting) and that the black lines are experimental data, both in Figure 6G.

The cases 1, 2, and 3 in Figure 6H that correspond to cases shown in 6F (now Figures 6H and G) shows examples of the three categories (non-persistence, normal persistence, and super-persistence). Note that there is no clear separation between these categories since there is a continuum of persistence rather than discreet classes. In the new figure 6H, the experimental black lines have improved visibility (we also show the SEM).

10) The physical model is very elegant despite being poorly and simply explained in a clear scheme that should be proposed in the Figure 6. Indeed, all the elements are accessible but widely spread in the main text and supplemental material. A scheme of this models and the different interplays involved will help its understanding to a wide readership.

We now have added a scheme at the beginning of Figure 6.